# Differential spatial computations in ventral and lateral face-selective regions are scaffolded by structural connections

Dawn Finzi [1✉], Jesse Gomez [2,3], Marisa Nordt [1], Alex A. Rezai[1], Sonia Poltoratski [1] & Kalanit Grill-Spector [1,2,4]

Face-processing occurs across ventral and lateral visual streams, which are involved in static and dynamic face perception, respectively. However, the nature of spatial computations across streams is unknown. Using functional MRI and population receptive field (pRF) mapping, we measured pRFs in face-selective regions. Results reveal that spatial computations by pRFs in ventral face-selective regions are concentrated around the center of gaze (fovea), but spatial computations in lateral face-selective regions extend peripherally. Diffusion MRI reveals that these differences are mirrored by a preponderance of white matter connections between ventral face-selective regions and foveal early visual cortex (EVC), while connections with lateral regions are distributed more uniformly across EVC eccentricities. These findings suggest a rethinking of spatial computations in face-selective regions, showing that they vary across ventral and lateral streams, and further propose that spatial computations in high-level regions are scaffolded by the fine-grain pattern of white matter connections from EVC.

[1] Department of Psychology, Stanford University, Stanford, CA, USA. [2] Neurosciences Program, Stanford University, Stanford, CA, USA. [3] Princeton Neuroscience Institute, Princeton University, Princeton, NJ, USA. [4] Wu Tsai Neurosciences Institute, Stanford University, Stanford, CA, USA. ✉email: dfinzi@stanford.edu

F ace perception is crucial for everyday social interactions and relies on brain computations across a series of face-selective regions in the occipital and temporal cortex. In humans, there are three visual processing streams[1,2], of which face-selective regions are contained in two[1,3–5]: the ventral and lateral streams[1]. The ventral face processing stream is thought to be involved in face recognition[6–9]. Cortically, it begins in early visual cortex (EVC: union of V1, V2, V3), continues to the inferior aspects of the occipital and temporal cortex, and contains several face-selective regions: one on the inferior occipital gyrus (IOG-faces, also referred to as the occipital face area, OFA), and two on the fusiform gyrus—one on the posterior fusiform (pFus-faces[10]) and one in the mid fusiform (mFus-faces[10])—which are collectively referred to as the fusiform face area (FFA[6]). These regions project[11] to an anterior temporal face patch[4]. The lateral stream is instead hypothesized to be a stream for dynamic[1,4,12,13], social[2,4,5,8], and multimodal[1,2,14] perception, as it processes transient[15] aspects of faces such as motion[12,13,16,17], expression[8,9], and gaze[18]. The lateral stream continues from early visual cortex to the superior temporal cortex. It consists of a region in the posterior superior temporal sulcus (pSTS-faces[3,10]) and a region on the main branch of the STS (mSTS-faces[3,10]), then projects to the anterior STS (aSTS-faces[10,11]). As the ventral and lateral face processing streams are optimized for different tasks, a central open question is: what are the basic computations separating these streams?

A basic characteristic of the visual system is the spatial computation by the receptive field[19] (RF), akin to a filter that processes visual information in a restricted part of visual space. As neurons with RFs that process similar parts of visual space are clustered in cortex, we can measure with fMRI the population receptive field (pRF)[20], which is the part of visual space processed by the collection of neurons in a voxel. In addition, how a cortical region spatially processes a stimulus depends on the way pRFs in the region tile the visual field[21], referred to as visual field coverage (VFC). While classic theories have hypothesized that pRFs are mainly a characteristic of early and intermediate visual areas, accumulating evidence suggests that pRFs are also a characteristic of high-level visual areas[20–23]. Therefore, we asked: what are the properties of pRFs and VFC in face-selective regions of the human ventral and lateral processing streams?

One possibility is that pRF properties and VFC are similar across face-selective regions of the ventral and lateral streams. A large body of research shows that people tend to fixate on faces[24–27]. This habitual fixation on faces has led researchers to propose eccentricity bias theory[28,29]. This theory is supported by findings that ventral face-selective regions respond more strongly to central than peripheral visual stimuli[28,29] and have a foveal bias—that is, denser coverage of the central than peripheral visual field[21,23]. In contrast, visual information related to other categories, such as places, which in the real world occupy the entire visual field, extends to the periphery of the visual field irrespective of fixation. Consistent with the predictions of eccentricity bias theory, place-selective regions are peripherally-biased[28]. Thus, eccentricity bias theory predicts that due to habitual fixation on faces, spatial computations in all face-selective regions, across both ventral and lateral processing streams, will be foveally-biased.

An alternative hypothesis predicts that pRF properties and VFC will be different across face-selective regions of the ventral and lateral streams because these streams are optimized for different tasks with different computational demands. Face recognition requires the fine spatial acuity afforded by central vision, predicting that ventral face-selective regions, which are involved in face recognition, will be foveally-biased[28]. However, social interactions often involve a group of people. As such, even when

fixating on one face, processing social aspects of multiple faces in the group may require peripheral vision[2]. Further, processing of dynamic information requires integrating optic flow across the visual field[30] and is faster in the periphery than in the fovea[15,31]. Thus, the computational demands hypothesis predicts that pRFs and VFC in lateral face-selective regions, which are involved in social and dynamic processing of faces, will extend to the periphery.

To test these hypotheses, we designed a pRF mapping experiment optimized to map pRFs in high-level visual regions (Fig. 1A), inspired by other experiments using complex stimuli including objects[32,33], faces[21], and words[22]. Using these data, we estimated pRFs in each voxel and then compared pRFs and VFC across face-selective regions in the ventral and lateral streams.

We also considered an important, related question: how do pRF characteristics and eccentricity biases emerge in high-level visual regions in the first place? A prevalent view suggests that the hierarchical organization of visual processing streams, as well as pooling operations from one stage to the next, generate successively larger pRFs in higher-level stages of the hierarchy. This hierarchical organization may be supported by sequential connections along regions constituting the processing stream[11,34–36]. However, other evidence shows that the human visual stream is not strictly hierarchical[2,34,37,38] as there are skip connections[35] between areas that are not consecutive in the hierarchy. For example, there is some evidence for direct white matter connections between EVC and face-selective regions of both ventral and lateral processing streams[34,37].

However, the nature of direct connections from early to later stages of the visual hierarchy and how they relate to pRFs in downstream regions remains a mystery. As EVC has a topographic representation of visual space, we examined the pattern of white matter connections between eccentricity bands in EVC and face-selective regions. We reasoned that to support large pRFs in high-level regions, there should be connections between a range of EVC eccentricities and face-selective regions. However, we hypothesized that one source of emergent spatial biases in face-selective regions could be an uneven distribution of connections stemming from central and peripheral eccentricities in EVC. Therefore, we predicted that foveally-biased regions would have a preponderance of white matter tracts emerging from central compared to peripheral eccentricities of EVC. In contrast, regions in which spatial computations extend to the periphery would have a more uniform distribution of connections across EVC eccentricities. We tested these predictions in a second experiment, in which we used diffusion MRI (dMRI) and fMRI in the same participants to determine the distribution of white matter tracts between EVC eccentricity bands and each of the face-selective regions.

## Results

**Toonotopy (retinotopy with cartoons) drives responses in the face-selective cortex more than standard checkerboard retinotopy.** To measure pRFs, 28 adults (14 female) participated in a wide-field pRF mapping experiment that used colorful cartoon images ("toonotopy", Fig. 1A). In the experiment, subjects fixated on a central dot and pressed a button when it changed color. As they performed this task, they viewed a bar extending 40° in length that systematically swept the visual field (Fig. 1A). Unlike standard pRF mapping[20,23] that uses flickering high contrast patterns, our bars contained random images from cartoons including colorful faces, objects, scenes, and text presented at a rate of 8 Hz. Faces in these cartoon images spanned the mapped visual field (Supplementary Fig. 1). In addition, subjects participated in a 10-category localizer

## (A) Experimental design

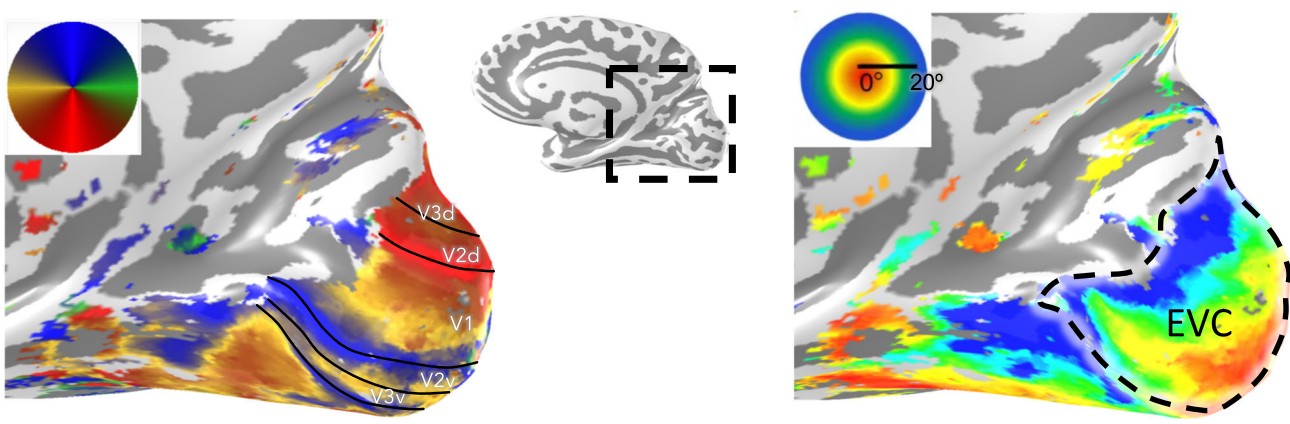

## (B) Polar angle and eccentricity maps, example subject

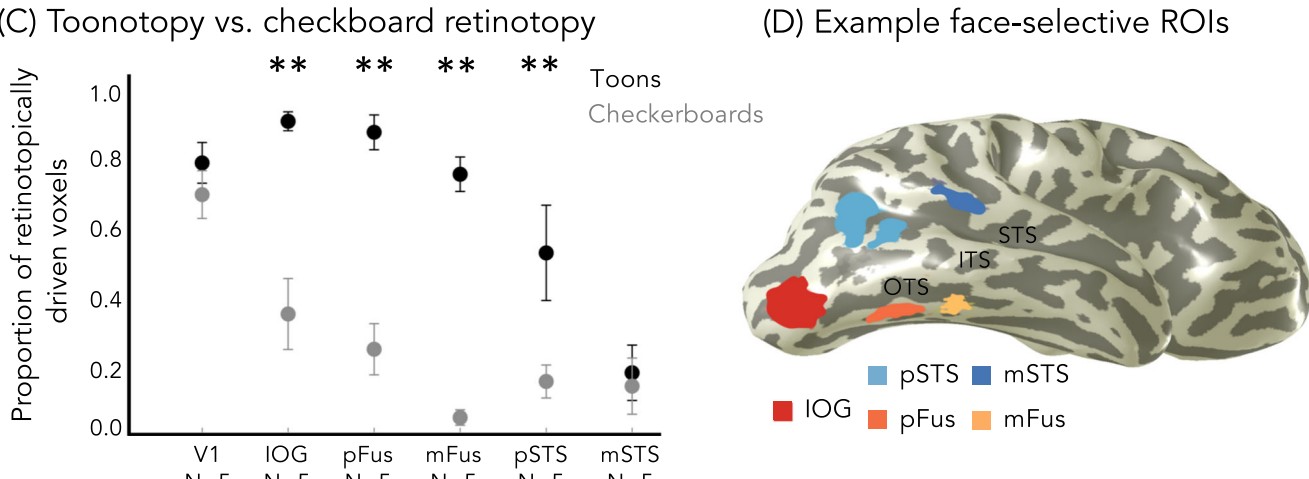

## (C) Toonotopy vs. checkboard retinotopy

## (D) Example face-selective ROIs

**Fig. 1 Toonotopy experiment and regions of interest. A** Experimental design. A bar containing colored cartoon images (similar to those shown here) changing at 8 Hz is swept across a gray background in 4 orientations (0°, 45°, 90°, 135°), each in 2 directions orthogonal to the bar. Each sweep takes 24 s, with blanks after six-bar steps (12 s) for each diagonal direction. Participants were instructed to fixate and indicate when the fixation dot changed color. **B** Example polar angle (left) and eccentricity (right) maps in a representative participant's inflated right hemisphere. Inset: zoomed medial view. Maps are thresholded at 20% variance explained, voxel level. Solid lines: boundary of early retinotopic areas. Dashed contour: early visual cortex: the union of V1, V2, and V3. **C** The proportion of voxels in each right hemisphere region in which pRF model explains >20% of their variance. Data averaged across five participants who underwent both checkerboard retinotopy and toonotopy. Error bars: ±SE (standard error of the mean). Asterisks: significant differences between toonotopy vs. checkerboard retinotopy; post-hoc Tukey $t$-tests (**$p < .01$, two-sided). IOG-faces: $t(44) = 4.98$, $p = 1.03 \times 10^{-5}$, $d = 0.75$; pFus-faces: $t(44) = 5.61$, $p = 1.28 \times 10^{-6}$, $d = 0.84$; mFus-faces: $t(44) = 6.28$, $p = 1.29 \times 10^{-7}$, $d = 0.95$; pSTS-faces: $t(44) = 3.32$, $p = .0018$, $d = 0.50$; mSTS-faces: $t(44) = 0.35$, $p = .73$, V1: $t(44) = 0.82$, $p = .42$. **D** Face-selective regions defined from the localizer experiment in a representative participant's inflated right hemisphere using contrast: faces vs. all eight other categories, $t > 3$, voxel level. Warm colors: ventral face-selective regions; Cold colors: lateral face-selective regions. Acronyms: IOG: inferior occipital gyrus; pFus: posterior fusiform. mFus: mid fusiform; OTS: occipito-temporal sulcus; ITS: inferior temporal sulcus; STS: superior temporal gyrus.

experiment[39] to independently define their face-selective regions.

From the localizer experiment, we defined three ventral face-selective regions (IOG, pFus, mFus-faces) and two lateral face-selective regions (pSTS, mSTS-faces) in each subject's brain (Fig. 1D, see "Methods" section). In addition, we defined in each participant their ventral place-selective region (CoS-places), which we include to replicate prior findings and as a control in certain analyses. All regions were found bilaterally in the majority of participants, except for mSTS-faces, which was lateralized to the right hemisphere in all but twelve participants. Due to the diminished number of mSTS-faces in the left hemisphere, as well as the vast literature reporting lateralization of face-selectivity[4,13,38], we analyze data from each hemisphere separately throughout this study, unless otherwise specified.

In the pRF mapping experiment, we used colorful cartoon stimuli and fast presentation rates in an attempt to maximally drive responses in the high-level visual cortex. To quantify the success of this approach, in five participants we compared toonotopy to a standard checkerboard pRF mapping experiment, which was identical to the toonotopy experiment except the bars contained black and white checkerboards that flickered at a rate of 2 Hz. In each participant and experiment, we fit a pRF for each voxel and identified the voxels for which the pRF model explained at least 20% of their variance.

Qualitatively, toonotopy pRF mapping yielded the typical retinotopic polar angle and eccentricity maps (Fig. 1B). Along the calcarine we found a hemifield representation, corresponding to V1, followed by mirror reversals of the polar angle (Fig. 1B-left). Likewise, we found the standard eccentricity map of the occipital lobe, with foveal representations close to the occipital pole, and systematically more peripheral eccentricities proceeding from posterior to anterior along the calcarine sulcus (Fig. 1B-right).

Quantitatively, we compared the proportion of retinotopically-modulated voxels across experiments. Differences between the ability of mapping stimuli to drive high-level face-selective regions are striking. Nearly 80% of mFus-faces voxels in the right hemisphere were driven in the toonotopy experiment, compared to less than 5% in the checkboard experiment (Fig. 1C, Supplementary Fig. 1-left hemisphere). As linear mixed models (LMMs) tolerate missing values[40] (e.g., when there are missing regions of interest (ROIs)), here and in subsequent analyses, we applied LMMs to analyze the data. The significance of fixed effects in these models was evaluated using analyses-of-variance with Satterthwaite approximations for degrees of freedom[41] (referred to as LMM ANOVA). A 2-way repeated-measures LMM ANOVA on the proportion of retinotopically-driven voxels with factors of the experiment (toonotopy/checkerboards) and ROI (V1/IOG/pFus/mFus/pSTS/mSTS) revealed a significant ROI × experiment interaction in both the right ($F(5,44) = 6.3$, $p = .00017$, $\eta_p^2 = 0.42$) and left hemispheres ($F(5,40) = 6.0$, $p = .00032$, $\eta_p^2 = 0.43$). This interaction reflects the pronounced effect the type of mapping experiment had on driving responses in most face-selective regions, particularly within the ventral stream, but not V1 or mSTS-faces. For the latter, the combined change in stimuli (cartoons vs. checkerboards) and presentation rate (8 Hz vs. 2 Hz) may have not been sufficient to drive neurons in the region, which prefer dynamic stimuli[13] (Fig. 1C, Supplementary Fig. 2).

**pRF centers in ventral face-selective ROIs are foveally-biased, while pRFs in lateral face-selective regions extend to the periphery.** Next, we compared pRF characteristics across ventral and lateral face-selective ROIs. Visualizing pRF centers across participants for each ROI (Fig. 2A) reveals differences across ventral

and lateral regions. pRF centers of ventral face-selective regions, IOG-faces, pFus-faces, and mFus-faces, are largely confined within the central 10° (Fig. 2, Supplementary Fig. 3). In contrast, pRF centers of lateral face-selective regions, pSTS-faces, and mSTS-faces, extend to the periphery, even past 30° from fixation. Notably, the distribution of pRF centers of pSTS-faces is more similar to CoS-places, a ventral place-selective region with a peripheral bias[29], than to ventral face-selective regions.

We quantified these observations by calculating the proportion of pRF centers within each of four eccentricity bands (0–5°/5–10°/10–20°/20–40°) for each participant and ROI (Fig. 2B). These eccentricities bands were chosen as they approximately occupy a similar cortical expanse due to cortical magnification[20,42]. As illustrated in Fig. 2B, the foveal bias is particularly striking in the right hemisphere, where ~70% or more of pRF centers in right ventral face-selective regions are located within the central 5° (right IOG-faces: mean ± SE = 0.68 ± 0.06; pFus-faces: 0.69 ± 0.05; mFus-faces: 0.80 ± 0.05), with few pRF centers located outside the central 10°.

Lateral face-selective ROIs show the opposite distribution of pRF centers across eccentricity bands. In these regions, the proportion of centers increases from central to peripheral eccentricity bands. In stark contrast to ventral face-selective ROIs, less than 20% of centers were located within the central 5° eccentricity band for lateral face-selective regions (right pSTS-faces: 0.19 ± 0.06; right mSTS-faces: 0.17 ± 0.07). Here, the majority of pRF centers are outside the central 10°.

A 2-way repeated-measures LMM ANOVA on the proportion of centers with eccentricity band (0–5°/5–10°/10–20°/20–40°) and stream (ventral: IOG/pFus/mFus and lateral: pSTS/mSTS) as factors revealed a significant eccentricity band × stream interaction in both hemispheres (right: $F(3,428) = 77.1$, $p < 2.2 \times 10^{-16}$, $\eta_p^2 = 0.35$; left: $F(3,376) = 41.0$, $p < 2.2 \times 10^{-16}$, $\eta_p^2 = 0.25$). Post-hoc Tukey's tests establish that this is driven by a significantly higher proportion of centers in the most foveal 0–5° eccentricity band in ventral vs. lateral face-selective regions (proportion higher in ventral than lateral–right: 0.54 ± 0.04, $t(428) = 12.2$, $p < 2.2 \times 10^{-16}$, $d = 0.59$; left: 0.41 ± 0.06, $t(376) = 7.1$, $p = 5.8 \times 10^{-12}$, $d = 0.37$), as well as a significantly lower proportion of centers for ventral vs. lateral regions in the two most peripheral eccentricity bands (proportion lower in ventral than lateral, 10–20° right: 0.32 ± 0.04, $t(428) = -7.4$, $p = 7.8 \times 10^{-13}$, $d = 0.36$; left: 0.30 ± 0.06, $t(376) = -5.3$, $p = 2.1 \times 10^{-7}$, $d = 0.27$; 20–40° right: 0.23 ± 0.04, $t(428) = -5.2$, $p = 3.6 \times 10^{-7}$, $d = 0.25$; left: 0.32 ± 0.06, $t(376) = -5.5$, $p = 5.7 \times 10^{-8}$, $d = 0.29$). These differences persist even if analyses are limited to the maximal extent of the stimulus (central 20°, Supplementary Fig. 4). In addition, differences in pRF center distributions are not a general characteristic of ventral vs. lateral regions, as CoS-places, which is a ventral region, has pRFs that extend into the far periphery (Fig. 2-green).

There were also significant differences in the distribution of pRF centers between hemispheres in ventral face-selective ROIs, revealed by a significant hemisphere by the band by ROI interaction ($F(6,544) = 8.0$, $p < 3.0 \times 10^{-8}$, $\eta_p^2 = 0.08$, 3-way repeated measures LMM ANOVA with factors of the hemisphere, eccentricity band, and ROI) and a significant hemisphere by band interaction ($F(3,544) = 21.0$, $p < 6.6 \times 10^{-13}$, $\eta_p^2 = 0.10$). As evident in Fig. 2B, pRF centers were concentrated more foveally in the right hemisphere than in the left, particularly in pFus-faces and mFus-faces. For both pFus-faces and mFus-faces, there were significantly more pRF centers in the 0°–5° band in the right than left hemisphere (pFus-faces-right: mean ± SE = 0.69 ± 0.05; left: 0.36 ± 0.06; post-hoc Tukey test $t(544) = 5.7$, $p = 2.3 \times 10^{-8}$, $d = 0.24$; mFus-faces-right: mean ± SE = 0.80 ± 0.05; left: 0.50 ± 0.07; post-hoc Tukey test $t(544) = 4.9$, $p = 1.0 \times 10^{-6}$, $d = 0.21$).

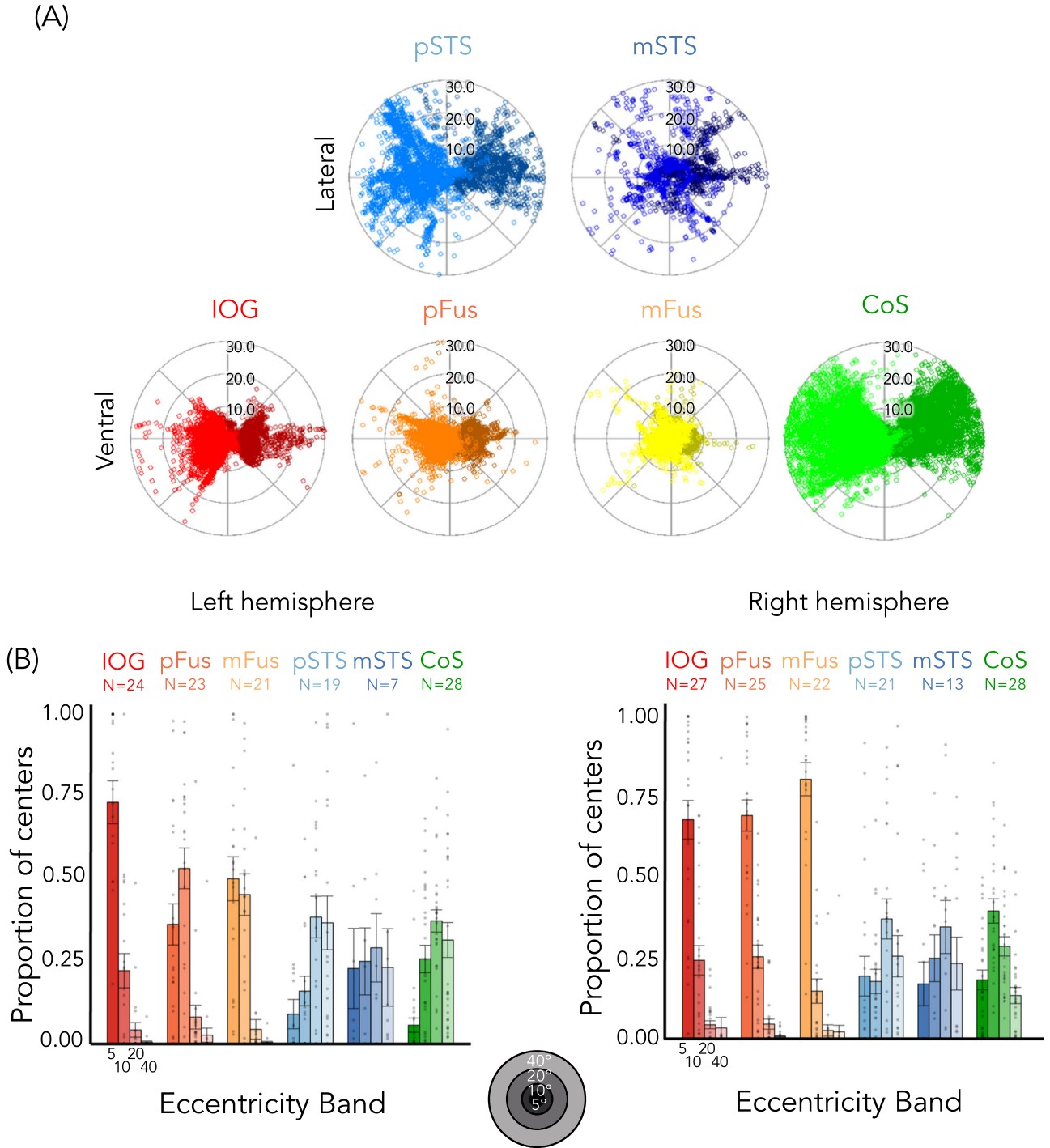

**Fig. 2 Population receptive field (pRF) centers in ventral face-selective ROIs have a foveal bias, while pRF centers in lateral face-selective regions (and CoS-places) extend to the periphery. A** Distribution of pRF centers; Each dot is a pRF center. Dark colors: Left hemisphere. **B** Bars: mean proportion of pRF centers of face-selective ROIs and CoS-places across eccentricity bands (0°–5°, 5°–10°, 10°–20°, and 20°–40°) averaged across participants (number indicated above each plot). Error bars: ±SE. Dots: individual participants. Warm colors: ventral face-selective regions; Cold colors: lateral face-selective regions. Acronyms: IOG: inferior occipital gyrus; pFus: posterior fusiform. mFus: mid fusiform; STS: superior temporal gyrus; CoS: collateral sulcus.

Together these analyses reveal: (i) differences in the distribution of pRF centers across face-selective regions of the lateral and ventral streams, and (ii) a higher foveal bias in right than left ventral face-selective regions.

**pRFs in lateral face-selective ROIs are larger than pRFs in ventral face-selective regions**. In addition to differences in pRF

locations, pRF sizes differ between lateral and ventral face-selective regions. Specifically, median pRF sizes are larger in lateral than ventral face-selective regions (Fig. 3A). In the right hemisphere, for example, the average across participants of median pRF sizes in pSTS-faces was 21.2° ± 1.6°, while in pFus-faces (the ventral region with the largest pRFs) the average was 9.7° ± 0.4°. To determine if these differences were significant, we calculated in each participant the median pRF size across ROIs in

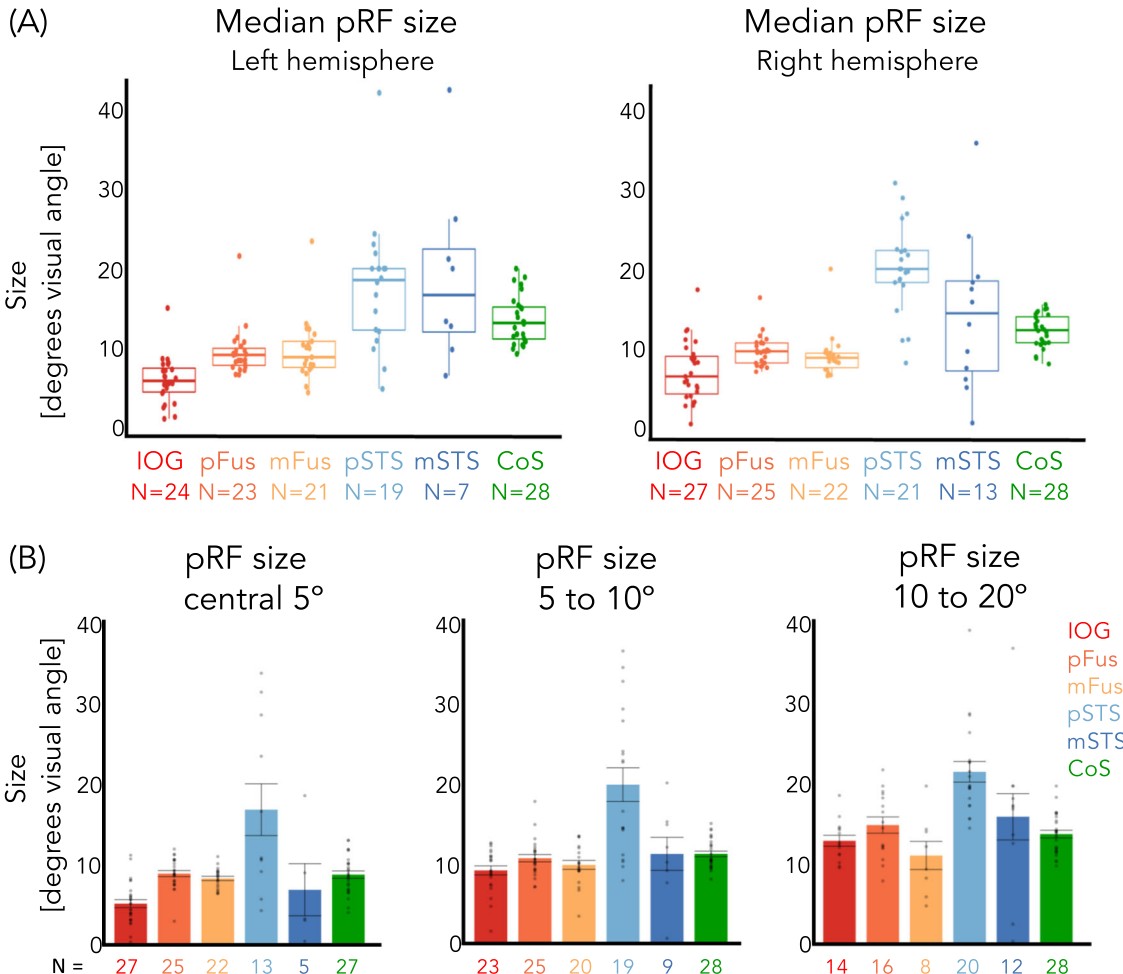

**Fig. 3 Population receptive fields (pRFs) in lateral face-selective regions are larger than in ventral face-selective regions. A** Median pRF size ($\frac{\sigma}{\sqrt{n}}$) in face-selective regions and CoS-places. Box: median, 25%, and 75% percentiles; lines: ±1.5 times interquartile range; Dots: individual participant median pRF size. **B** Average median pRF size across participants for each eccentricity band of right hemisphere regions. Left hemisphere ROIs were not included as we could only identify in 6 participants any voxels whose pRF eccentricity was less than 5° in any of the lateral ROIs. Dots: Individual participant median pRF size. Error bars: ±SE. The number of subjects per ROI and eccentricity band is indicated under each bar. Warm colors: ventral face-selective regions; Cold colors: lateral face-selective regions. Acronyms: IOG: inferior occipital gyrus; pFus: posterior fusiform. mFus: mid fusiform; STS: superior temporal gyrus; CoS: collateral sulcus.

each stream (ventral: IOG/pFus/mFus and lateral: pSTS/mSTS) and then compared values across streams. Results show that in both hemispheres pRFs were significantly larger in lateral than ventral face-selective regions (paired $t$-tests; right: $t(24) = -4.3$, $p = .00025$, $d = 0.88$; left: $t(22) = -6.1$, $p = 3.6 \times 10^{-6}$, $d = 1.31$). Differences between ROIs were significant (1-way repeated measures LMM ANOVAs on median pRF size, right ROIs: IOG/ pFus/mFus/pSTS/mSTS, $F(4,86) = 13.6$, $p = 1.3 \times 10^{-8}$, $\eta_p^2 = 0.39$; left ROIs: IOG/pFus/mFus/pSTS/mSTS, $F(4,74) = 22.0$, $p = 5.4 \times 10^{-12}$, $\eta_p^2 = 0.54$), and were driven by significant differences between pSTS-faces and each of the ventral face-selective regions (post-hoc Tukey tests, all $ts > 4.8$, $ps < = .0001$, Supplementary Table 1), as well as mSTS-faces and each of the ventral face-selective regions (post-hoc Tukey tests, all $ts > = 3.5$, $ps < = .0064$, Supplementary Table 1).

As pRF size increases with eccentricity in retinotopic regions[20], we next tested if differences in pRF sizes across pSTS and ventral regions were simply driven by the higher number of peripheral pRFs in the former than the latter. We reasoned that if that were the case, then the comparison of pRF sizes within the same eccentricity band should reveal no difference across ROIs. However, comparison of pRF sizes within the same eccentricity

band revealed significant differences between median pRF across ROIs (Figs. 3B, 1-way LMM ANOVAs with the factor of ROI, right hemisphere, 0–5° band: $F(4,87) = 11.8$, $p = 1.0 \times 10^{-7}$, $\eta_p^2 = 0.35$; 5–10° band: $F(4,72) = 6.3$, $p = .00020$, $\eta_p^2 = 0.26$). Moreover, across both the 0–5° and 5–10° bands, pRFs were significantly larger in right pSTS-faces than in any other region except mSTS-faces (all $ts > 3.4$, $ps < = .0085$, Supplementary Table 2).

**Ventral face-selective ROIs have dense coverage of the central visual field, while lateral face-selective ROIs have coverage that extends to the periphery.** How do differences in pRF locations and sizes of face-selective ROIs affect each ROI's visual field coverage (VFC)? VFC was calculated as the proportion of pRFs covering each point in the visual field, for each participant and ROI, then averaged across subjects. Results reveal three main findings illustrated in Fig. 4A: (1) In all ROIs, VFC is concentrated in the contralateral visual field (Supplementary Fig. 5A, B, quantification of contra/ipsi index), (2) ventral face-selective ROIs have dense coverage concentrated around the center of the visual field, and (3) lateral face-selective ROIs have more diffuse coverage that extends further into the periphery than ventral face-selective

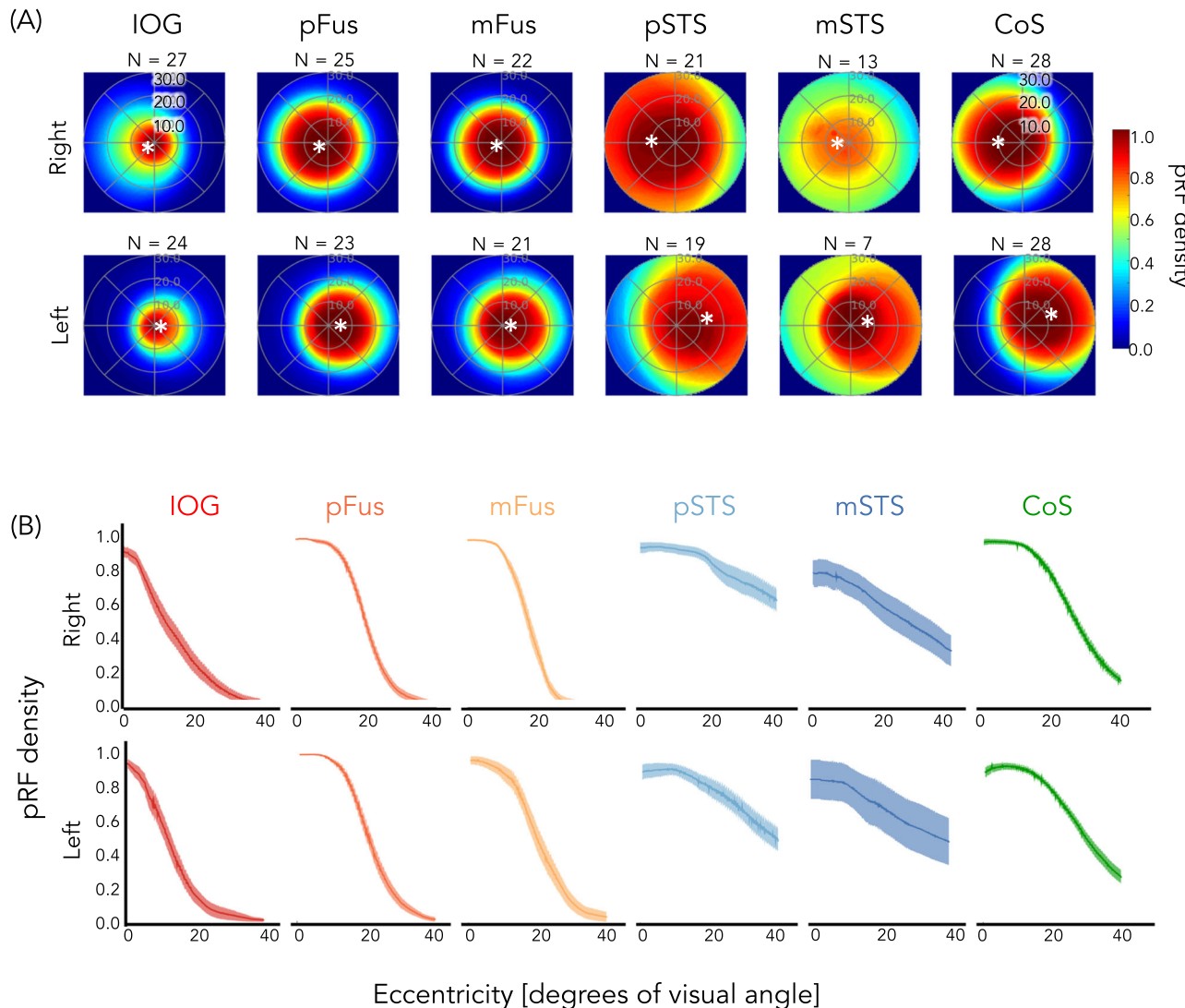

**Fig. 4 Dense coverage of the central visual field in ventral face-selective regions, but coverage extending to the periphery in lateral face-selective regions (and CoS-places). A** Average visual field coverage of each ROI across participants (number indicated above each plot). Visual field coverage is computed as the proportion of population receptive fields (pRFs) covering each point in the visual field for each participant and then averaged across participants. Asterisk: average location of the center of mass of all pRF centers in each ROI. **B** Average pRF density as a function of eccentricity for the contralateral visual field of each ROI. Data were calculated per participant and then averaged across participants. Shaded area: ±SE. Warm colors: ventral face-selective regions; Cold colors: lateral face-selective regions. Acronyms: IOG: inferior occipital gyrus; pFus: posterior fusiform. mFus: mid fusiform; STS: superior temporal gyrus; CoS: collateral sulcus.

ROIs. These differences are not a general characteristic of ventral vs. lateral regions, as ventral CoS-places exhibits VFC that extends into the periphery. Additionally, we find no significant evidence for either a lower or upper visual field bias in any ROI, except for a lower visual field bias in right IOG-faces and an upper visual field bias in left CoS-places (Supplementary Fig. 5C).

To quantify differences in VFC across face-selective regions, we calculated the average pRF density as a function of eccentricity for the contralateral visual field of each ROI. Ventral face-selective regions displayed high pRF density close to the fovea that decreased sharply beyond ~10°, while pRF density in lateral face-selective regions decreased more moderately with increasing eccentricity (Fig. 4B). We summarized the relationship between pRF density and eccentricity by fitting both generalized logistic and linear functions for each participant's pRF density curve per ROI as the ventral regions appear to be best approximated by a logistic function but the lateral face-selective regions by a linear function (see "Methods" section). Irrespective of the model-fitting

approach, we find significant differences in the fitted parameters between ventral and lateral face-selective regions. For the linear model, we examined the slope of the line as negative slopes indicate higher pRF density near the fovea than the periphery, and slopes close to 0 indicate similar pRF densities across eccentricities. Results reveal (i) significant differences between the average slopes of ventral and lateral face-selective regions (paired t-tests; right: $t(24) = -9.8$, $p = 7.4 \times 10^{-10}$, $d = 2.00$; left: $t(22) = -8.2$, $p = 3.7 \times 10^{-8}$, $d = 1.75$), whereby slopes for ventral face-selective ROIs were more negative than for lateral face-selective ROIs and (ii) significant differences between the average slopes of individual face-selective ROIs (right: $F(4,85) = 47.6$, $p < 2.2 \times 10^{-16}$, $\eta_p^2 = 0.69$; left: $F(4,84) = 26.9$, $p = 2.3 \times 10^{-14}$, $\eta_p^2 = 0.56$, 1-way repeated measures LMM ANOVAs on the slopes with factor ROI). Specifically, slopes in lateral face-selective regions—pSTS-faces and mSTS-faces—were significantly closer to zero than any of the ventral face-selective regions (all $ts > 4.1$, $ps < = .0008$, post-hoc Tukey tests, Supplementary Table 3).

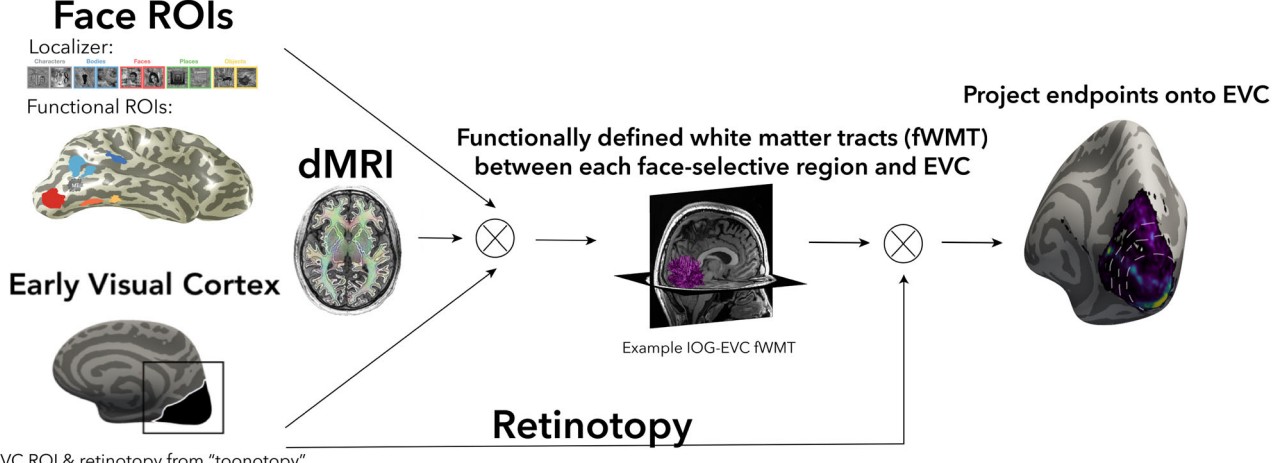

**Fig. 5 Functionally-defined white matter tracts (fWMT) between functional ROIs and early visual cortex (EVC).** We combined dMRI and fMRI in each participant to determine fWMT between each functional ROI and EVC. From left to right, fMRI: from the localizer experiment, we defined in each participant their face-selective ROIs; from the toonotopy experiment, we defined in each subject their EVC and eccentricity map. dMRI: using MRTrix3 with ACT[43] we defined the whole-brain connectome of each participant, which was culled to remove false alarm tracts with Linear Fascicle Evaluation[68]. fWMT of each ROI: all tracts that intersect with both the functional ROI and EVC. Endpoints in EVC: We projected the end points of each fWMT in EVC and related them to the eccentricity map. All analyses were done in each participant's native brain space.

In addition, bilateral pFus-faces and right mFus-faces had significantly more negative slopes than IOG-faces (all $ts < -3.3$, $ps < =.012$, Supplementary Table 3), indicating that the former ROIs have a larger foveal bias than the latter. Similarly, the parameters for both the inflection point and the lower asymptote of the fitted logistic function were significantly different between ventral and lateral regions in both hemispheres, such that ventral face-selective regions had smaller valued lower asymptotes (paired $t$-tests; right: $t(24) = -4.6$, $p = 0.00011$, $d = .94$; left: $t(22) = -4.2$, $p = .00041$, $d = .89$) and inflection points (paired t-tests; right: $t(24) = -4.0$, $p = 0.00055$, $d = .81$; left: $t(22) = -2.3$, $p = 0.034$, $d = .48$) than lateral face-selective regions.

**Face-selective regions have direct white matter connections to the early visual cortex.** Given our findings of differences in pRF properties and visual field coverage across ventral vs. lateral face-selective regions, we asked how white matter connections from early visual cortex (EVC) contribute to this differentiation. We reasoned that the foveal bias in ventral face-selective regions may arise in part from a preponderance of connections from foveal eccentricities of EVC. In contrast, the more uniform tiling of the visual field by pRFs in lateral face-selective regions may arise from a more uniform pattern of white matter connections from EVC eccentricities to these face-selective ROIs along the STS.

To test these predictions, we acquired an additional diffusion-weighted MRI (dMRI) scan from 16 of our participants (see "Methods" section). We combined dMRI data with the fMRI data from both the localizer and toonotopy experiments to identify the functionally-defined white matter tracts (fWMT) that connect each face-selective region with EVC (Fig. 5). We used anatomically-constrained tractography (ACT)[43], which uses the gray-white matter interface to seed the tractography. This guarantees that the resulting fiber tracts reach the gray matter, which is crucial for testing our hypothesis. As a control, we also defined the fWMT that connects CoS-places with EVC. We asked: (1) Are there white matter tracts that connect each of the face-selective ROIs and EVC? (2) If so, how are they distributed across EVC eccentricities?

We found direct white matter connections between each face-selective and place-selective region and EVC in all participants

where we could localize the region (Fig. 6A, representative subjects; all subjects, Supplementary Fig. 6). Qualitatively, fWMT between face-selective regions and EVC were largely consistent across participants, although the proportion of tracts varied substantially by region (Fig. 6B). IOG-faces had the highest percentage of its tracts connecting to EVC (right: mean ± SE = 37.7 ± 4.2%, left: 39.6 ± 6.4%), followed by pFus-faces (right: 13.2 ± 1.7%, left: 14.7 ± 2.8%). mSTS-faces had the lowest percentage, with less than 3% of all mSTS-faces tracts connecting to EVC (right: 2.0 ± 0.5%; left: 2.3 ± 0.8%). Due to both the low number of left mSTS-faces ROIs and the low percentage of fibers connecting to EVC, we excluded left hemisphere mSTS-faces from subsequent analyses. However, in both hemispheres, ventral mFus-faces and lateral pSTS-faces had similar percentages of tracts to EVC (right mFus-faces: 9.5 ± 1.6%, left mFus-faces: 8.4 ±0.9%; right pSTS-faces: 10.8 ± 1.7%, left pSTS-faces: 9.0 ± 1.6%), suggesting that differences in fWMT to EVC do not stem from differences in our ability to identify tracts across ventral and lateral streams. In addition, despite their adjacent locations, ventral regions mFus-faces and CoS-places have different percentages of connections to EVC. Specifically, CoS-places (right: 22.0 ± 2.9%; left: 17.4 ± 3.1%) has twice the percentage of connections to EVC as mFus-faces. These data further underscore that factors other than anatomical location determine the percentage of tracts between functional ROIs and EVC.

**Ventral and lateral face-selective regions differentially connect to EVC eccentricities.** To test whether ventral and lateral face-selective regions are differentially connected to EVC eccentricities, we evaluated to which eccentricity band in EVC these tracts connect. Eccentricity values were determined from each participant's toonotopy data. We then quantified what proportion of tracts from the category-selective ROI to EVC connected to each of the four EVC eccentricity bands (Fig. 7).

For ventral face-selective regions in both hemispheres, the highest proportion of white matter endpoints were located within the central 5° of EVC, while the fewest endpoints were within the most peripheral EVC eccentricity band, mirroring the results for pRF centers (Fig. 7). In fact, the majority (>66%) of the tracts connecting EVC to ventral face-selective ROIs originate within the central 10°. This overrepresentation of tract endpoints within

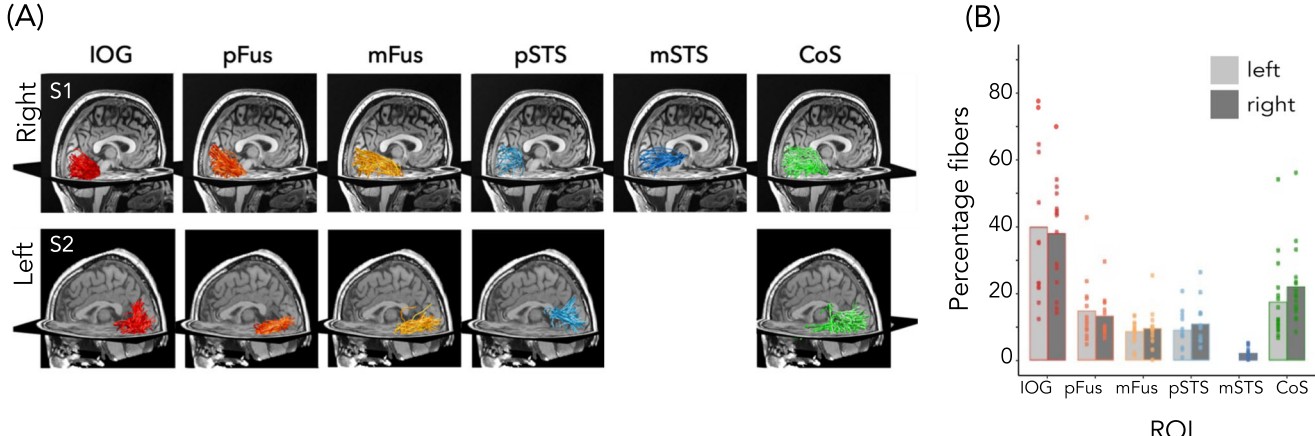

**Fig. 6 White matter tracts connecting EVC and face-selective regions (and CoS-places). A** Example fWMT between each face-selective region and EVC in example participants. Top: right hemisphere, Subject 1; Bottom: left hemisphere, Subject 2. **B** Mean percentage of fWMT between each functional ROI and EVC. 100% indicates that all tracts starting at an ROI connect to EVC and 0 indicates that no tracts connect to EVC. Percentage calculated for each participant and then averaged across participants. Left hemisphere-IOG-faces: $N = 13$, pFus-faces: $N = 13$, mFus-faces: $N = 14$, pSTS-faces: $N = 12$, CoS-places: $N = 16$. Right hemisphere-IOG-faces: $N = 15$, pFus-faces: $N = 13$, mFus-faces: $N = 13$, pSTS-faces: $N = 14$, mSTS-faces: $N = 14$, CoS-places: $N = 16$. Dots: Individual participant percentage. Lighter bars: left hemisphere; Darker bars: right hemisphere. Warm colors: ventral face-selective regions; Cold colors: lateral face-selective regions. Acronyms: IOG: inferior occipital gyrus; pFus: posterior fusiform. mFus: mid fusiform; STS: superior temporal gyrus; CoS: collateral sulcus.

central EVC is not the case for lateral face-selective regions. In these regions, tract endpoints are instead more evenly distributed across EVC eccentricity bands, with at least 20% of endpoints in the most peripheral eccentricity band (>40°, Fig. 7). Importantly, this difference is not due to a general bias that favors connections from foveal eccentricities in EVC to VTC, as the tracts that connect ventral CoS-places to EVC are more uniformly distributed across EVC eccentricities. In fact, nearly half of the tracts from EVC to CoS-places (right: 48%, left: 43%) originate in eccentricities ≥20°.

To test the significance of endpoint distributions in EVC between streams, we ran a 2-way repeated-measures LMM ANOVA on the proportion of tract endpoints in EVC with factors of eccentricity band and stream. We found a significant eccentricity band × stream interaction in both hemispheres (right: $F(3,264) = 7.9$, $p = 4.5 \times 10^{-5}$, $\eta_p^2 = 0.08$; left: $F(3200) = 19.5$, $p = 3.8 \times 10^{-11}$, $\eta_p^2 = 0.23$). As with the proportion of pRF centers, post-hoc Tukey's tests establish that stream differences in both hemispheres are driven by a significantly higher proportion of tract endpoints in the most central eccentricity band (0–5°) in ventral vs. lateral face-selective regions (proportion higher in ventral than lateral, right: $0.12 \pm 0.03$, $t(264) = 3.8$, $p = .0002$, $d = 0.23$; left: $0.20 \pm 0.04$, $t(200) = 5.0$ $p = 1.3 \times 10^{-6}$, $d = 0.35$), as well as a significantly lower proportion of centers for ventral vs. lateral regions in the most peripheral eccentricity band (proportion lower in ventral than lateral, 20–40°, right: $0.09 \pm 0.03$, $t(264) = -2.8$, $p = .0056$, $d = 0.17$; left: $0.16 \pm 0.04$, $t(200) = -4.0$, $p = .0001$, $d = 0.29$). To control for any effects of ROI size, we repeated this analysis with 5 mm disk ROIs, created at the center of each functionally defined region (Supplementary Fig. 7). As in the main analysis, a 2-way repeated-measures ANOVA on the proportion of fiber endpoints in EVC confirms significant eccentricity band × stream interactions in both hemispheres (right: $F(3256) = 8.7$, $p = 1.65 \times 10^{-5}$, $\eta_p^2 = 0.09$; left: $F(3,200) = 17.3$, $p = 5.248 \times 10^{-10}$, $\eta_p^2 = 0.21$).

Together, we find that the ventral and lateral face-selective regions differentially connect to EVC. Connections from ventral face-selective regions are biased to the central 10° in EVC, while connections from lateral face-selective regions are distributed across eccentricity bands and extend into the far periphery. White

matter connectivity differences between ventral and lateral face-selective ROIs mirror the different distributions of pRF centers, suggesting that white matter connections may provide a scaffolding for differential pRF properties and VFC between streams.

## Discussion

Using a multimodal approach, we tested how different face-selective regions of visual cortex represent visual space and whether this can be traced to white matter connections with EVC. We find that (i) contrary to the prevailing view, not all face-selective regions are foveally biased, as lateral face-selective regions show pRFs and VFC extending into the periphery, and that (ii) these differences are reflected in differential patterns of white matter connections with EVC, as lateral face-selective regions exhibit white matter connections that are more evenly distributed across eccentricities than ventral face-selective regions.

Consistent with prior research illustrating a foveal bias in face-selective regions[21,29,44], our data show that pRFs of ventral face-selective regions process information around the center of gaze. Surprisingly, we find that this is not a general property of face-selective regions: pRFs of lateral face-selective regions are both more peripheral and larger than those of the ventral stream. This finding is consistent with the prediction of the computational demands hypothesis and requires a rethinking of spatial computations in face perception. Furthermore, these properties do not appear to simply reflect a general difference between ventral and lateral visual cortex, as pRFs and white matter connections of CoS-places, which is in ventral temporal cortex, more closely resemble those of the lateral face-selective regions. However, an interesting question for future research is whether these differences across face-selective areas in the ventral and lateral streams extend to body-selective regions, which neighbor the face-selective regions in both streams[1,45].

A substantial body of research[3–5] highlights differences in the computational goals of ventral and lateral streams: ventral face-selective regions are thought to be involved in face recognition, while lateral regions are thought to be involved in dynamic[4,12,13]

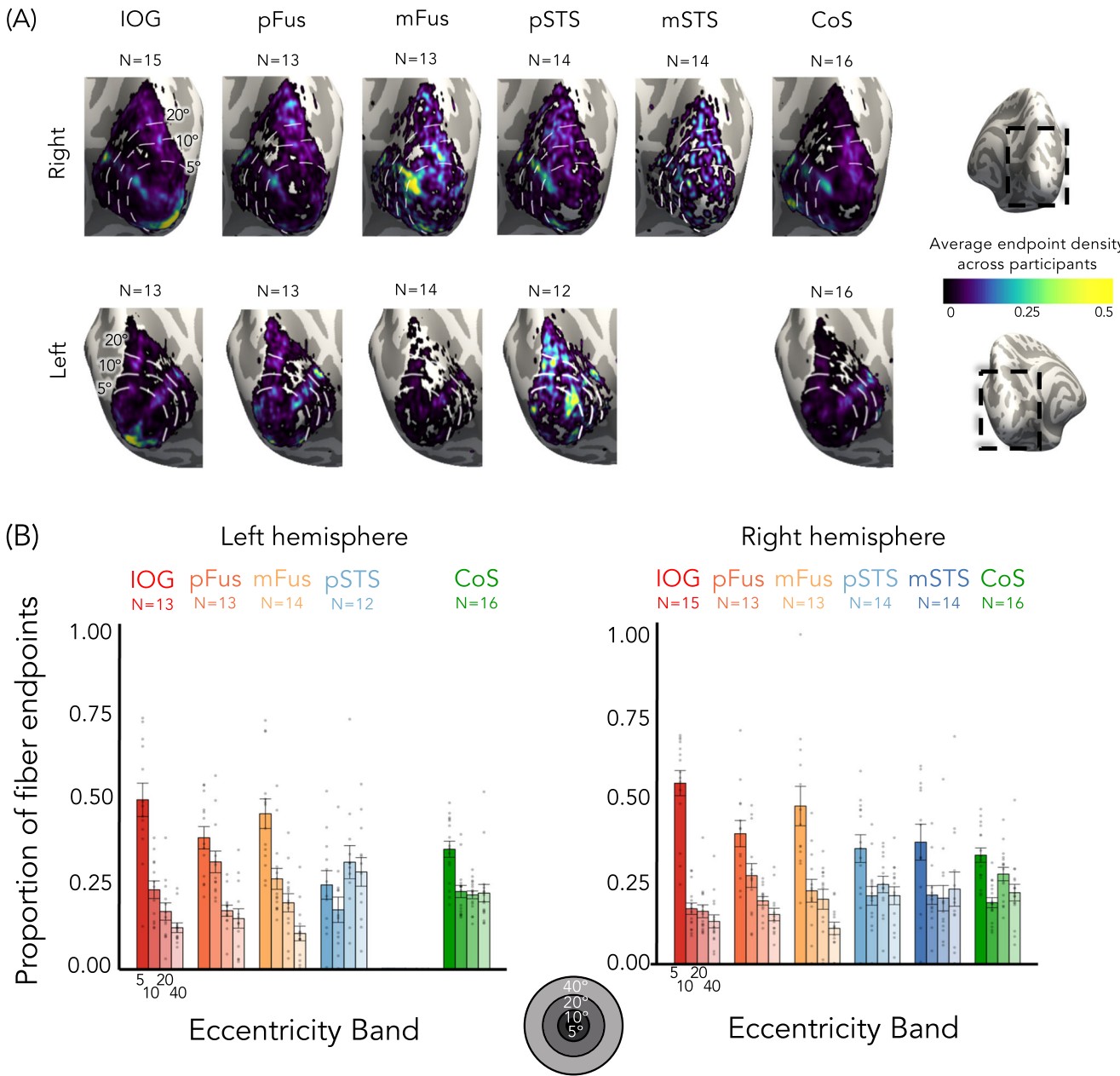

**Fig. 7 Tracts from lateral and ventral face-selective regions have a different distribution across early visual cortex (EVC) eccentricities. A** Mean endpoint density of fiber tracts connecting each ROI with EVC shown on the zoomed medial inflated cortical surface of the FreeSurfer average brain; Top: right hemisphere; Bottom: left hemisphere. Color map: average density across participants. Eccentricity bands (white dotted lines) were derived from the average of all participants' retinotopies. **B** The average proportion of tract endpoints from each ROI that terminate in each of four EVC eccentricity bands (0°–5°; 5°–10°; 10°–20°; 20°–40°). Analyses were done in each participant's native brain space. Bars: mean across participants; Each dot is a participant; Error bars: ±SE. Warm colors: ventral face-selective regions; Cold colors: lateral face-selective regions. Acronyms: IOG: inferior occipital gyrus; pFus: posterior fusiform. mFus: mid fusiform; STS: superior temporal gyrus; CoS: collateral sulcus.

and social aspects of face perception, such as expression[8,9] and gaze[18,46]. What might the function be of the differential pRF properties across these streams? The foveal bias of ventral face-selective regions has been attributed to the high visual acuity necessary for face recognition, which is achieved by foveal vision[28]. Indeed, people tend to fixate on the center of the face during recognition[23,27,28], putting the VFC of ventral face-selective regions on the internal face features that carry information about identity[23].

Our finding of pRFs extending to the periphery in lateral face-selective regions raises an additional question: what is the benefit of these peripheral computations? We propose several non-

mutually exclusive hypotheses that can be tested in future research. One possibility is that peripheral computations in lateral face-selective regions and their direct white matter connections to peripheral EVC may facilitate rapid processing of dynamic[16,47] and transient[15] face information, as the speed of visual processing is faster in the periphery than the fovea[31]. Another possibility is that large and peripheral pRFs in lateral face-selective regions may enable integrating information across the entire visual field, which is important for computing shape and motion information from optical flow[30] across the entire face or whole person[48]. A third possibility is that in social situations involving a group of people, peripheral pRFs in lateral face-selective regions may allow

for inferring social information from faces in the periphery, which are not directly fixated upon, such as an eye-roll or an encouraging smile. A behavioral prediction from our data, as well as other prior research[49], which can be tested in the future, is that performance on tasks related to the lateral stream (e.g., judging facial expressions) will decline less as a function of eccentricity than tasks related to the ventral stream (e.g., judging facial identity).

Though dMRI is not capable of resolving whether the connections we find are feedforward or feedback, the majority of connections in the cortex are likely reciprocal[50]. Thus, we conjecture that included in the white matter connections we discovered are tracts that originate in EVC and project to face-selective regions. In addition, we do not intend to suggest that all differences in spatial computations between ventral and lateral face-selective regions can be attributed to direct connections with EVC. It is likely that hierarchical connections throughout the ventral stream[36] also contribute to shaping pRFs in face-selective regions, though we are not able to resolve these shorter connections with the present dMRI resolution. Nonetheless, together with hierarchical connections, small differences in the distribution of connections from EVC eccentricities to face-selective regions may result in pronounced differences in spatial computations by pRFs across processing streams.

Our finding that visual field properties in face-selective regions are reflected in differential white matter connections with EVC supports the idea that structural connections serve a functional role. Our data suggest that not only do face-selective regions in different streams process visual input differently, they actually receive different visual input to begin with. It is also interesting that the foveal bias of white matter connections between EVC and ventral face-selective regions resembles the foveal bias of functional connections to EVC[51–53], which are present in infancy[51]. An important question for future research is whether white matter connections between EVC and downstream regions constrain where face-selective regions emerge during development[54].

Our results not only explicate the nature of connections between EVC and face-selective regions[34,37] but also substantially advance the precision with which we can understand connections within the human brain. In contrast to many dMRI studies that examine white matter connections of entire fascicles or between brain regions, to our knowledge, this is the first study that identifies white matter connections associated with a topographic map in individual human brains. As receptive fields, topographic maps, and processing streams are key computational characteristics of sensory (e.g., audition[55]) and cognitive (e.g., numerosity[56]) brain systems and across species[44], our innovative approach can be applied to study how topographic computations emerge and are scaffolded by white matter connections in many brain systems and species.

In sum, we discovered differential pRF and VFC characteristics across face-selective regions in the ventral and lateral processing streams, which are mirrored by differences in white matter connections with eccentricity bands in EVC. These findings suggest a rethinking of computations in face-selective regions, and further, propose that spatial computations in high-level regions may be scaffolded by the pattern of white matter connections from EVC.

## Methods

**Participants**. Twenty-eight participants (14 female, ages 22 to 45, mean age 26.8 ± 5.1 years) from Stanford University participated in the study, sixteen of whom returned for diffusion MRI (dMRI). All participants had normal or corrected-to-normal vision. Participants gave written informed consent, and all procedures were approved by the Stanford Internal Review Board on Human Subjects Research.

**MRI data acquisition**. Participants were scanned using a General Electric Discovery MR750 3T scanner located in the Center for Cognitive and Neurobiological Imaging (CNI) at Stanford University. A phase-array 32-channel head coil was used for all data acquisition except for the retinotopic mapping experiment (16-channel head coil).

Anatomical scans: for each of 21 participants we obtained a whole-brain, anatomical volume using a T1-weighted BRAVO pulse sequence (resolution: 1 × 1 × 1 mm, TI = 450 ms, flip angle: 12°, 1 NEX, FoV: 240 mm). In 7 participants we used quantitative MRI measurements using the protocols from[57] to generate a 1 × 1 × 1 mm, T1-weighted anatomical image of their brain. Anatomical images of each participant's brain were used for gray/white matter tissue segmentation, which was then used for cortical surface reconstruction and visualization of data on the cortical surface.

fMRI category localizer experiment: A category localizer was run to identify voxels that responded more strongly to one category vs. other categories of stimuli in order to localize face-selective and place-selective regions of interest (ROIs). In the experiment, participants were presented with grayscale images of stimuli from five domains, each with two categories (faces: child, adult; bodies: whole, limbs; places: corridors, houses; characters: pseudowords, numbers; objects: cars, guitars[39]). Images from each category were presented in 4 s trials at a rate of 2 Hz, intermixed with 4 s blank trials. Each category was shown eight times per run in counterbalanced order. Participants were instructed to fixate on a central point and perform an oddball detection task, identifying 0–2 randomly presented phase scrambled images within a trial. Runs were 5 min and 18 s long and each participant completed 3 runs with different images and category order. Data were collected with a simultaneous multi-slice EPI sequence with a multiplexing factor of 3 to acquire near whole-brain (48 slices) volumes at TR = 1 s, TE = 30 ms at a resolution of 2.4 mm isotropic. Data were acquired with a one-shot T2*-sensitive gradient echo sequence and slices were aligned parallel to the parieto-occipital sulcus.

pRF mapping experiment: Each participant completed four runs of a wide-field pRF mapping experiment to model the part of the visual field that elicits a response from each voxel (i.e., each voxel's pRF). In each run, participants fixated on a central stimulus and were required to press a button whenever the central fixation dot changed color as bars were swept across the screen (mean accuracy = 96.0 ± 6.3%). Due to our interest in higher-level category-selective regions, bars of width 5.7° swept across high-contrast, colorful, cartoon stimuli that spanned a wide-field circular aperture (40° × 40° of visual angle). Images randomly changed at a rate of 8 Hz. The bars randomly revealed a portion of the cartoon images and they largely include object parts rather than entire objects (Fig. 1A). We did not include video clips in our mapping stimuli as we wanted to ensure that participants maintained fixation (which is critical for pRF mapping) instead of tracking motion in videos. To better drive the lateral stream regions, which prefer moving vs. stationary stimuli[4,12,13,15–17] we presented our stimuli at a higher rate of 8 Hz, as prior studies have shown that neurons in MT and in other lateral stream areas that prefer motion also prefer higher frequency stimuli (with peak sensitivity around 10 Hz[39,58]).

In each run, bars of four orientations (0°, 45°, 90°, 135°) swept the visual field in eight directions (2 opposite directions orthogonal to the bar orientation) with blanks (mean luminance gray background with a fixation) interspersed at regular intervals. Each run lasted 3 min and 24 s. Eye-tracking was not conducted in the scanner, but visual field coverage of early visual field maps (V1–V3) shows the expected hemifields and quarterfields (Fig. 1B for an example), indicating that participants maintained stable fixation.

Diffusion magnetic resonance imaging (dMRI): In 16 of the participants, we acquired whole-brain diffusion-weighted, dual-spin echo sequences (60 slices, TE = 96.8 ms, TR = 8000 ms, 96 diffusion directions, b0 = 2000 s/mm$^2$) at a resolution of 2.0 × 2.0 × 2.0 mm. Ten non-diffusion-weighted images were collected at the beginning of each scan.

**Preprocessing and data analysis**. Anatomical data: T1-weighted images were automatically segmented using FreeSurfer 5.3 and then manually validated and corrected using ITKGray. Using these segmentations, we generated cortical surfaces in FreeSurfer.

fMRI data analysis: fMRI data analysis was performed in MATLAB and using the mrVista analysis software developed at Stanford University. All data were analyzed within individual participants' native brain anatomy space without spatial smoothing. Data were motion-corrected within and between scans using mrVista motion correction algorithms, and then manually aligned to the anatomical volume. The manual alignment was optimized using robust multiresolution alignment.

Definition of V1–V3 (EVC): V1, V2, and V3 were defined in all participants using their individual data from the pRF mapping experiment. Maps of pRF phase and eccentricity were projected onto an inflated cortical surface reconstruction for each participant and borders between the retinotopic maps were drawn on this cortical surface. The boundary was defined as the center of polar angle reversal occurring at the vertical or horizontal meridian[23]. V2 and V3 were drawn as quarterfields separated by V1 and were later combined to produce a hemifield representation. The early visual cortex (EVC) ROI was then defined as the union of V1, V2, and V3. We chose this particular grouping for two main reasons: (i) V1, V2, and V3 share an eccentricity representation and are often grouped together in the hierarchy of the visual stream, separate from hV4, which is classified as an

intermediate area, and (ii) we were concerned that seeding for tractography may be too narrow if limited to one retinotopic area. Thus, we generated an EVC ROI that is the union of V1, V2, V3 to ensure that we had a full and accurate representation of the distribution of streamlines from eccentricity bands in early visual cortex.

*Definition of face-selective and place-selective functional regions of interest:* The functional ROIs were defined in individual participants from the category localizer experiment using anatomical and functional criteria[39]. Statistical contrasts of faces > all eight other stimuli (for face-selective ROIs) and places > all eight other stimuli (for place-selective ROIs) were thresholded at *t*-values >3, voxel level, in each participant, as in previous work[23,39]. Face-selective ROIs were defined in the inferior occipital gyrus (IOG-faces, right hemisphere: $N = 27$; left hemisphere: $N = 24$), posterior fusiform gyrus (pFus-faces, right: $N = 25$; left: $N = 23$), mid fusiform gyrus (mFus-faces; right: $N = 23$; left: $N = 25$), posterior superior temporal sulcus (pSTS-faces, right: $N = 25$; left: $N = 23$), and medial superior temporal sulcus (mSTS-faces, right: $N = 24$; left: $N = 12$). As prior studies[59] suggest that mSTS-faces is larger for moving stimuli than for still stimuli, such as those we used in our localizer, we sought to match the size of the mSTS-faces in our study to that reported in other studies by placing a 1 cm radius disk ROI centered on the functional ROI. We placed the disk on the center of the ROI as prior studies[60] indicate that this is a stable characteristic of functional ROIs. Analyses comparing this disk ROI and the original functional ROI based on the *t*-value threshold can be found in Supplementary Fig. 8. The same pattern of results is found for the disk and functional mSTS-faces ROIs, but the former allows us to include more data for mSTS-faces.

An additional place-selective ROI along the collateral sulcus was defined by contrasting responses to places vs. other stimuli ($t > 3$, voxel-level, CoS-places, right: $N = 28$; left: $N = 28$). This ROI corresponds to the parahippocampal place area[61].

*Estimating pRFs:* We modeled the population receptive field (pRF) in each voxel using the compressive spatial summation (CSS) model[62]. Time-course data were transformed from the functional slices to the T1-weighted anatomy using trilinear interpolation. We estimated pRFs within each gray matter voxel. The pRF model fits a 2-dimensional Gaussian for each voxel, with parameters *x* and *y* describing the location of the center of the pRF, and *σ*, the standard deviation of the Gaussian in degrees of visual angle. An additional compressive summation exponent (*n*) is fit for each voxel to capture the nonlinear summation properties of pRFs in later stages of the visual hierarchy[21,62]. We define the size of the pRF as: $\frac{\sigma}{\sqrt{n}}$ [62]. Candidate pRFs are estimated and a predicted time-series is produced by convolving the product of the binarized stimulus sequence and the candidate pRF with an HRF. The model parameters *x*, *y*, and *σ*, are then iteratively adjusted to minimize the sum of squared errors between the predicted time-series and the observed time-series. Phase ($\operatorname{atan}(\frac{y}{x})$) and eccentricity ($\sqrt{x^2 + y^2}$) of each voxel are derived from the *x* and *y* coordinates of the pRF center to generate phase and eccentricity maps, respectively. Voxels were included for subsequent analysis if the variance explained by the pRF model was greater than 20%. In addition, to ensure accurate pRF fits by the optimization procedure, voxels whose *σ* remained at the initial minimum value (0.21°) were excluded from further analysis. For analyses of pRFs in each ROI, we include data from participants for whom the ROI was identified and at least 10 voxels (corresponding to a volume of ≥10 mm³) of the ROI were retinotopically modulated according to the above criteria (variance explained ≥20%, pRF size >0.21°). The number of participants per ROI is as follows: IOG-faces, right hemisphere $N = 27$, left $N = 24$; pFus-faces, right $N = 25$, left $N = 23$; mFus-faces, right $N = 22$, left $N = 21$; pSTS-faces, right $N = 21$, left $N = 19$; mSTS-faces, right $N = 13$, left $N = 7$; CoS-places, right $N = 28$, left $N = 28$. To ensure that the results do not depend on our variance explained threshold, we repeated all analyses with a 10% variance explained threshold and the results remain the same (Supplementary Notes).

*Visual field coverage (VFC):* VFC is defined as the portion of the visual field that is processed by the set of pRFs spanning the ROI. To calculate the VFC for a given ROI and participant, all voxels in an ROI that passed the above inclusion criteria are included. Each pRF is modeled with a binary circular mask centered at the pRF center and with a diameter of $\frac{2\sigma}{\sqrt{n}}$. For each location in the visual field, the VFC value is the pRF density at that location. Group average VFC maps are created by averaging the individual VFC maps for that ROI. Thus, the group VFC depicts the mean pRF density at each location averaged across subjects. To assess differences in VFC across ROIs, we fitted both linear and generalized logistic functions relating pRF density to eccentricity in each ROI and participant, and then compared the parameters (across participants) of different ROIs in each stream. The linear function was of the form $y = ax + b$, where *a* represents the slope and *b* the y-intercept. The generalized logistic function was of the form: $= a + \frac{b-a}{1+10^{(c-x) \cdot d}}$, where fitted parameters *a*, *b*, *c*, and *d* represented the lower asymptote, the upper asymptote, the inflection point, and the steepness of the curve, respectively. Mean adjusted $R^2$ was high for the linear model (mean ± SE adjusted $R^2$ across ROIs and subjects: right = .81 ± .01, left = .80 ± .02) and was, unsurprisingly given the additional number of parameters, even higher for the generalized logistic function (mean ± SE adjusted $R^2$ across ROIs and subjects: right = .97 ± .01, left = .96 ± .02).

For each functional ROI, we also calculated indices of contra-laterality vs. ipsi-laterality and upper vs. lower visual field bias of the VFC (Supplementary Fig. 5). Laterality of VFC was calculated for each participant and ROI as the mean coverage in the contralateral visual field minus the mean coverage in the ipsilateral visual

field, divided by the sum of the two $\left(\frac{contra - ipsi}{contra + ipsi}\right)$. Similarly, the upper field bias for each participant and ROI was quantified as the average coverage in the contralateral upper visual field minus the average coverage in the contralateral lower visual field, divided by the sum of the two $\left(\frac{upper - lower}{upper + lower}\right)$.

*dMRI data analysis:* dMRI data were preprocessed using a combination of tools from MRtrix3[63], ANTs, and FSL, as in[64,65]. We denoised the data using: (i) a principal component analysis, (ii) Rician based denoising, and (iii) Gibbs ringing corrections[66]. We also corrected for eddy currents and motion using FSL and performed bias correction using ANTs. The dMRI data were then registered to the average of the non-diffusion-weighted images and aligned to the corresponding anatomical brain volume of the participant using rigid body transformation. We used constrained spherical deconvolution[63] with up to eight spherical harmonics (lmax = 8) to calculate the voxel-wise tract orientation distributions (FOD). These FODs were then used for tractography.

*Tractography:* Ensemble tractography[67] with linear fascicle evaluation (LiFE[68]) was performed using the preprocessed dMRI data and consisted of 3 main steps:

1. We used MRtrix3[63] to generate five candidate connectomes which varied in the maximum angle (2.9°, 5.7°, 11.5°, 23.1°, and 47.2°). Each candidate connectome was seeded on the gray-white matter interface (GWMI) using anatomically constrained tractography (ACT[43]) and consisted of 500,000 streamlines. This approach allows us to generate multiple connectomes with different degrees of curvature, as opposed to restricting our estimations to a single connectome with one particular set of parameters[67]. Each candidate connectome was generated using probabilistic tract tracking with the following parameters: algorithm: IFOD2, step size: 1 mm, minimum length: 4 mm, maximum length: 200 mm. ACT[43] uses the GWMI from the FreeSurfer segmentation of each participant's brain to seed the tracking algorithm. This enabled us to focus only on those fiber tracts that reach the gray matter.

2. The five candidate connectomes were concatenated into one ensemble connectome containing a total of 2,500,000 streamlines.

3. We used linear fascicle evaluation (LiFE[68]) to validate the ensemble connectome. LiFE was used to determine which fiber estimates make a significant contribution to predicting the dMRI data and remove false alarm tracts that did not significantly contribute to predicting the dMRI data.

*Functionally defined white matter tracts (FDWT):* To determine the white matter tracts that are associated with each functional ROI, we intersected the tracts with the GWMI directly adjacent to each functional ROI for each participant. This yielded functionally defined white matter tracts (FDWT) for each face and place-selective ROI.

*EVC-fROI connections:* To identify tracts connecting the category-selective regions and early visual cortex, we intersected the FDWT of each ROI with the GWMI underneath EVC (union of V1, V2, V3), and then used these pairwise tracts for individual subject visualization and endpoint density estimation.

*Quantification of endpoint density:* We used MRtrix3[63] and track-density imaging (TDI) to create tract maps of the streamline endpoints of the FDWT for each functional ROI. These endpoint density maps were transformed to the cortical surface using FreeSurfer. We examined the portion of the endpoint density maps that intersected EVC and determined the corresponding eccentricity values. Eccentricity was derived from the pRF mapping experiment and each participant's individual retinotopic maps. We used the track-density values to weigh each eccentricity value based on the density of endpoints connecting to that eccentricity. We then quantified the proportion of endpoints in each of four eccentricity bands (0° to 5°, 5° to 10°, 10° to 20°, 20° to 40°) for each functional ROI and participant separately. For mSTS-faces, we used the 1 cm disk ROI centered on the functional ROI, using the same ROI per subject across pRF mapping and dMRI experiments.

*Size control:* As functional ROIs vary in size across participants and regions, we performed a control analysis to evaluate the endpoint density using constant-size disk ROIs (radius = 5 mm) centered on each participant's functional ROIs. The goal of this analysis was to test whether differences across ROIs could be attributed to differences in ROI size. Results of these analyses are presented in the Supplementary Materials (Supplementary Fig. 7).

**Statistical analyses.** Statistical analyses were computed using the lmerTest[41], lsmeans, and effectsize packages in R (https://cran.r-project.org/package=lmerTest, https://cran.r-project.org/package=lsmeans, https://cran.r-project.org/package=effectsize). Linear mixed-effects models (LMMs) were used for statistical analyses because our data include missing data points (not all participants have all ROIs). LMMs were constructed separately for each hemisphere, with one exception: to test the effect of hemisphere on the proportion of pRF centers by eccentricity band in the ventral face-selective ROIs (i.e., to test the lateralization of the foveal bias), we constructed an LMM which can be expressed as: proportion of pRF centers ~ eccentricity band + ROI + hemisphere + (eccentricity band*ROI) + (ROI × hemisphere) + (eccentricity band × hemisphere) + (eccentricity band × ROI × hemisphere) + (1|subject). The significance of LMM model terms was evaluated using repeated-measures analyses-of-variance (LMM ANOVAs; type III), with Satterthwaite's method of correction for degrees of

freedom[40]. Post-hoc tests were done by computing Tukey's Honest Significant Difference (HSD) where the family-wise confidence level is set at 0.95. p-values reported from these post-hoc tests are thus corrected for multiple comparisons. All statistical tests conducted in the manuscript followed this procedure other than four paired *t*-tests. One paired *t*-test was conducted to compare the average pRF size across streams (average pRF size across ventral ROIs vs. average pRF size across lateral ROIs), one was conducted to compare slopes of the pRF density by eccentricity lines across streams (average slope across ventral ROIs vs. average slope across lateral ROIs), and two were conducted to compare parameters of the generalized logistic function fit to the pRF density by eccentricity data across streams: (i) average lower asymptote across ventral ROIs vs. average lower asymptote across lateral ROIs and (ii) average inflection point across ventral ROIs vs. average inflection point across lateral ROIs.

## Data availability

Individual nifti files are available upon request. Source data are provided with this paper.

## Code availability

fMRI data were analyzed using the open source mrVista software package (available on GitHub: http://github.com/vistalab). T1-weighted anatomicals for 7 subjects were generated using the mrQ software package (https://github.com/mezera/mrQ). dMRI data were analyzed using mrTrix3 (http://www.mrtrix.org/). Custom code (all available on GitHub) was used for dMRI preprocessing (https://github.com/vistalab/RTP-preproc; https://github.com/vistalab/RTP-pipeline) and processing of the pRF experiment (D.F. and K.G.-S. VPNL/Toonotopy: Initial release. (2021). https://doi.org/10.5281/zenodo.4560632). Code for all further analysis, including reproducing all figures and statistics, can be found at https://doi.org/10.5281/zenodo.4560752 (Finzi, D., Gomez, J., Nordt, M., Rezai, A. A., Poltoratski, S., and Grill-Spector, K, Differential spatial computations in ventral and lateral face-selective regions are scaffolded by structural connections, VPNL/fibeRFs, 2021, https://doi.org/10.5281/zenodo.4560752).

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

## Acknowledgements

This work was supported by a William R. and Sara Hart Kimball Stanford Graduate Fellowship awarded to D.F., NIH grants (grant numbers RO1EY02231801, RO1EY02391501 to K.G.S.), Ruth L. Kirschstein National Research Service Award (grant number F31EY027201 to J.G.), the NSF Graduate Research Development Program (grant number DGE-114747 to J.G.), and a fellowship of the German National Academic Foundation (grant number NO 1448/1-1 to M.N.).

## Author contributions

D.F. designed the experiment, collected data, analyzed the data, and wrote the manuscript. J.G. collected the data, contributed to data analysis, and contributed to the manuscript. M.N., and A.A.R., collected the data and contributed to the manuscript. S.P. contributed to data analysis and contributed to the manuscript. K.G.-S. designed the experiment, contributed to the data analysis, and wrote the manuscript.

## Competing interests

The authors declare no competing interests.
