## [Peer Review File · Nature Communications]

Reviewer #1 (Remarks to the Author):

The authors probe differences in spatial computation in the ventral and lateral stream face areas by looking at population receptive field pRF mapping and white matter connectivity. They show that ventral face selective regions are driven from the center of gaze, and that lateral regions sample in a way that extends peripherally. Data from diffusion MRI between early visual cortex (EVC) and ventral and lateral face areas provide converging evidence for the pRF mapping in showing a preponderance of connections between foveal EVC and ventral areas, and more uniformly-sampled EVC connections to lateral areas.

Evaluation

It has long been long known that the organization of ventral cortex references an eccentricity bias with respect to retinal input (Levy, 2001; Hasson et al. 2002 and Weiner et al. 2014). This earlier literature references functional neuroimaging methods and links between anatomy and function in making that case. In this study, the authors focus on regions within the face processing system and extend from ventral to lateral face areas to make an analogous argument. They do this with direct measures of pRF mapping and with white matter connectivity data. This is elegant work that places the face system (perhaps) into that broader organizing principle—with some caveats. The work employs state-of-the art methods and the findings are well-supported quantitatively. Also, the combination of the pRF mapping and the converging white matter data makes a strong and specific case. The data here are valuable and will contribute to the literature and improve our understanding of how these areas collaborate with each other and with early visual areas in face processing.

I have some general concerns about the way the hypotheses are framed and a few specific technical concerns. I also have some comments on the Discussion.

1.) Introduction – (lines 47-48) That statement holds for low level vision, but there is some question as to whether it makes sense in high-level visual areas. There is a standing tension in the literature about the degree to which the organization of high level visual cortex is influenced by “retinal space”. Eccentricity is clearly an organizing principle in high-level visual cortex, but it is one of many—so this statement may be too strong.

2.) (lines 55-76) I realize this is a short-form paper, and it does not give the authors much space to lay out hypotheses—but I did not find the formulation of the hypotheses for Exp. 1 entirely clear. Two hypotheses are pitted against each other. Eccentricity bias – which would predict that the fovea would be important for all face processing regions (ventral and lateral)- due to “habitual fixation on faces”. The other hypothesis assumes foveal centering for ventral regions, but a need for fast, motion-based computation in the lateral regions, which presumably draw on more peripheral resources (stated later but not clear at this point in the manuscript). I did not see how this fits the general framework of general eccentricity bias (progressing from ventral to lateral) from previous work. Is the first hypothesis a straw man or does the inclusion of STS make it a different question? Also, the inference that motion draws on more peripherally extended resources should be made explicit at this point in the paper.

3.) The “toonotopy” method seems clearly a better approach than the checkerboard methods for driving the ventral face areas. I would assume, however, that this reflects a combination response

based on what something is (face/person), and where it is on the retina. So, it is perhaps not surprising that overall more high-level area voxels are driven by the cartoons. It would be useful to know the extent to which the toons used in the pRF mapping show faces centered near the fixation point. That might account for at least some foveal bias in the more ventral areas—though it would not account for the more even distribution of pRF centers in the lateral areas.

4.) I was surprised that mSTS responses were like V1 on this comparison (Fig 1 C), rather than the other face areas, including pSTS. (line 196). Any thoughts on why this might be? Given that the toons were static—is it possible that the form part of the stimulus did not drive voxels as much as it may have for the ventral face areas?

5.) Discussion – lines 468 -478 – I am not entirely convinced by “speed” account of why lateral areas have more peripherally extended input. It seems possible that any kind of coherent motion computation would require a larger area of the face (i.e., typical face motions would likely not “fit” in the fovea, but would instead involve the entire face and head). (Also, the speed of expression processing in humans is likely also due in part to fast, simultaneous subcortical processing.)

6.) A final general comment is the use of the term “spatial processing” throughout, which I found confusing. Arguably, lateral (dorsal stream) areas contribute far more to spatial orienting, etc. and may require a more complete representation of ego-centric space than ventral regions. This might account for more complete EVC space representations. The term spatial processing for lateral areas makes sense. It is not clear what “spatial processing” means in ventral areas. Arguably, in natural viewing, foveation is used to examine face features (eyes, mouth, etc.) in more detail. It is possible that the advantage of foveal input to ventral areas is that it may be the most efficient place to find information useful for categorization. It does not necessarily follow that the function of these ventral areas is to better represent that part of visual space. This seems to be way the term is used in the paper. This may be splitting hairs, but I think that the term, as applied here to ventral and lateral areas, is not meant in entirely the same way.

Reviewer #2 (Remarks to the Author):

This work, led by Dawn Finzi of the the Grill-Spector lab, documents some interesting differences between lateral and ventral face-selective regions of the human brain. The principal finding is that face-selective regions in the superior temporal sulcus (lateral stream) are biased toward more peripheral regions of the visual field, with coarser spatial sampling (larger pRFs), compared to face areas on the ventral stream. This result is demonstrated clearly in a series of analyses, and complemented by the interesting anatomical finding that fiber tracts, derived from diffusion weighted imaging, preferentially connect foveal regions of V1, V2, and V3 to ventral face areas (matching the measured spatial selectivity) whereas more peripheral regions of V1, V2, and V3 are preferentially connected to lateral face areas. The work nicely brings together findings from three important areas in human visual neuroscience; category-selectivity in cortex, retinotopy, and anatomical pathways. It may serve as a useful benchmark for many other studies. For example, by

describing systematic differences between lateral and ventral face area properties, the authors will help constrain questions of homology between human face areas and those of other species. This work will likely also lead to interesting followups on how the lateral and the ventral face areas communicate with one another during tasks requiring the functions of both streams. In short, the clear set of findings on an important topic make this an impactful paper. I find most of the analyses clear and convincing, and as such I have only minor concerns.

Minor

1. Eccentricity analyses (e.g., figure 2B) tend to go out to 40 deg. The mapping stimulus only went out to 20 deg. While it is possible to effectively map a pRF whose center is outside the maximal stimulus extent, the precision of the eccentricity estimates drops. It might be more appropriate to bin data beyond some threshold (say 20 deg, or maybe 25 deg) rather than to assume the eccentricity can be measured accurately all the way out to 40 deg. Another possibility would be to limit voxels to those whose center is within some criterion of the stimulus extent dependent on pRF size (say, within 1 STD). At a minimum, the authors should consider whether uncertainty at these high eccentricities could substantially affect any key results or conclusions. (I would think probably not!)

2. Density vs eccentricity. The density vs eccentricity functions of different areas comprise a key result (e.g., Figure 4b). The plots themselves are pretty convincing, but summarizing the distributions by linear fits is probably not optimal. The data look like they might be better described by a sigmoid of some kind, or perhaps piecewise linear (limited to a small number of pieces). Of course a line has the advantage of interpretation - one can summarize a single parameter (slope). Nonetheless, it looks a line is a poor fit to some of the distributions, esp the ventral face areas in Fig 4B.

3. The authors' estimates of pRF centers go out to 40 deg, and they find a fairly large number of such voxels in lateral face areas. How do the authors reconcile this with the fact that, as they note in several places in the text, faces are typically foveated? A face would need to be unusually close for it to extend out to 40 deg when foveated. Perhaps the point is that even though faces are *often* foveated, they are not *always* foveated? Or perhaps these areas play a role in choosing face targets in the periphery to foveate? In any case, it would help to comment on this issue. Simply indicating that these areas process dynamic information and that dynamic information is represented differently in the periphery does not by itself explain why it would be useful to have peripheral face representations.

Very minor.

1. The authors often refer to the ventral stream as being involved in perception of static properties. I think this language can be improved and clarified. While faces may be static during fMRI experiments, they are presumably rarely entirely static in natural viewing. Perhaps the authors mean "stable" properties such as identity that do not change over time. The point is, while a property might be stable, a face typically is not (entirely) static. Static vs dynamic face perception sounds more like a description of an experimental paradigm than the functional selectivity of a brain area.

2. On p 2 (line 47) the authors state that spatial processing is the basic computation of the visual system. Perhaps. Others might argue that color, depth, motion, etc are equally "basic". Perhaps change to "A basic" ...?

3. Throughout the text, the authors refer to EVC, which they define as V1, V2, and V3. It is true that

many authors sometimes do this. Nonetheless, a few sentences of justification explaining why these areas are similar to each other and why they are different from other areas would help motivate the grouping. This might come from cluster analysis in published work, or just evidence that these areas have the clearest retinotopic maps, or that they show relatively similar response patterns, etc.

4. P. 115 the authors refer to standard retinotopy experiments as containing flickering checkerboards. Perhaps some do, but the most common ones I know of (eg Dumoulin and Wandell, 2008) tend to have drifting contrast patterns rather than flickering. Moreover the patterns are not always checkerboards, but often dartboards or other achromatic contrast patterns (such as Kay et al 2013, JNP). Hence “contrast patterns” would probably be more appropriate than “checkerboards”.

5. Novelty of toonotopy. It is fine for the authors to describe the toonotopy as a novel design, since this specific design is novel. Nonetheless, there have been many retotopy studies testing patterns other than simple contrast patterns, including Point-Light Walkers (Saygin, Sereno); words (Le ... Wandell, 2017), the HCP retinotopy experiments (Benson et al 2018); natural movies like Buffy the Vampire Slayer (Sereno, though perhaps only in abstract form). In most of these cases, the stimulus was chosen for similar reasons to this paper, namely to get better activation in higher level visual areas. I think it would be appropriate to cite some of these, perhaps whichever came earliest.

6. Line 361 is missing a reference. It literally says “{ref}”.

Signed,
Jon Winawer
NYU

Reviewer #3 (Remarks to the Author):

Finzi and colleagues examined the pRF and white matter connections in the ventral and lateral face-selective areas. They examined the hypothesis that pRF and connections from EVC are foveal in the ventral stream (consistent with previous reports) whereas the lateral face areas represent more peripheral visual field information related to their role in processing dynamic facial information.

The paper is very well written and the analyses are well performed and presented. Based on the title and the abstract, though, I was expecting to see much clearer findings than those that are actually reported. The results of the ventral face areas are foveally biased as expected and was found previously, but the lateral areas do not show as consistent picture as described. The pSTS and mSTS are often showing different patterns of results. Also, the mSTS is found in a very small number of participants (6 out of 21) and it is therefore hard to draw conclusions on the entire lateral face-stream based on such small sample size. It also makes the statistical analysis more problematic when an interaction with ROI is examined for such a small number of subjects in one of the ROIs relative to the others.

This could have been resolved if the regions would have been localized with a dynamic face localizer. The STS face areas – in particular the more anterior ones that were hardly found in the current study - show twice as much response to dynamic than static faces, and the size of the regions responding to dynamic face stimuli in the STS is much larger. However, face-selective areas were defined with static images of faces and non-face stimuli, which generated face-selective activations in the ventral

face areas as expected, whereas activations of the lateral face areas were less consistent, in particular, in the mSTS. Also the pSTS, that is typically found also for static faces, is expected to be much larger for dynamic faces than the one that is found for static faces.

Given that the goal of the study was to assess differences between lateral and ventral face regions, and in particular differences that may derive from the different roles that the two streams play in the processing of dynamic and static face stimuli – the regions that were defined may not be suitable for testing these hypotheses. Indeed, results in the mSTS are not always consistent with the hypothesis and the pSTS and mSTS sometimes generate different patterns of responses (e.g. Figure 3; Figure 7).

This is very unfortunate because the authors do have the capability and expertise to perform this work well. However, based on the current data, conclusions on the different computations of the two face streams based on foveal vs peripheral visual processing is not well supported as could have been.

Point by point answers to reviewer comments are indicated in blue below each comment.

Reviewer #1 (Remarks to the Author):

The authors probe differences in spatial computation in the ventral and lateral stream face areas by looking at population receptive field pRF mapping and white matter connectivity. They show that ventral face-selective regions are driven from the center of gaze, and that lateral regions sample in a way that extends peripherally. Data from diffusion MRI between early visual cortex (EVC) and ventral and lateral face areas provide converging evidence for the pRF mapping in showing a preponderance of connections between foveal EVC and ventral areas, and more uniformly-sampled EVC connections to lateral areas.

Evaluation

It has long been long known that the organization of ventral cortex references an eccentricity bias with respect to retinal input (Levy, 2001; Hasson et al. 2002 and Weiner et al. 2014). This earlier literature references functional neuroimaging methods and links between anatomy and function in making that case. In this study, the authors focus on regions within the face processing system and extend from ventral to lateral face areas to make an analogous argument. They do this with direct measures of pRF mapping and with white matter connectivity data. This is elegant work that places the face system (perhaps) into that broader organizing principle—with some caveats. The work employs state-of-the art methods and the findings are well-supported quantitatively. Also, the combination of the pRF mapping and the converging white matter data makes a strong and specific case. The data here are valuable and will contribute to the literature and improve our understanding of how these areas collaborate with each other and with early visual areas in face processing.

We thank the reviewer for their enthusiastic summary of our study.

I have some general concerns about the way the hypotheses are framed and a few specific technical concerns. I also have some comments on the Discussion.

1.) Introduction – (lines 47-48) That statement holds for low level vision, but there is some question as to whether it makes sense in high-level visual areas. There is a standing tension in the literature about the degree to which the organization of high level visual cortex is influenced by “retinal space”. Eccentricity is clearly an organizing principle in high-level visual cortex, but it is one of many—so this statement may be too strong.

This is a good point and similar to a point that was raised by Reviewer #2.

Action: We have clarified these sentences, on pg. 2 we now write: “A basic characteristic of processing in the visual system is computations by receptive fields.... While classic theories have hypothesized that RFs are mainly a characteristic of early and intermediate visual areas, accumulating evidence suggests that pRFs are also a characteristic of high-level visual areas²⁰⁻

²³. Therefore, we asked: what are the properties of pRFs and VFC in face-selective regions of the human ventral and lateral processing streams?”

2.) (lines 55-76) I realize this is a short-form paper, and it does not give the authors much space to lay out hypotheses—but I did not find the formulation of the hypotheses for Exp. 1 entirely clear. Two hypotheses are pitted against each other. Eccentricity bias – which would predict that the fovea would be important for all face processing regions (ventral and lateral)- due to “habitual fixation on faces”. The other hypothesis assumes foveal centering for ventral regions, but a need for fast, motion-based computation in the lateral regions, which presumably draw on more peripheral resources (stated later but not clear at this point in the manuscript). I did not see how this fits the general framework of general eccentricity bias (progressing from ventral to lateral) from previous work. Is the first hypothesis a straw man or does the inclusion of STS make it a different question? Also, the inference that motion draws on more peripherally extended resources should be made explicit at this point in the paper.

We thank the reviewer for highlighting the need for better formulation of the hypotheses.

Action: We have revised the introduction to expand on the justification and evidence supporting the two competing hypotheses.

On pg. 3, we write: *One possibility is that pRF properties and VFC are similar across face-selective regions of the ventral and lateral streams. A large body of research shows that people tend to fixate on faces²⁴⁻²⁷. This habitual fixation on faces has led researchers to propose Eccentricity Bias Theory^{28,29}. This theory is supported by findings that ventral face-selective regions respond more strongly to central than peripheral visual stimuli^{28,29}, and have a foveal bias – that is, denser coverage of the central than peripheral visual field^{21,23}. In contrast, visual information related to other categories, such as places which in the real world occupy the entire visual field, extends to the periphery of the visual field irrespective of fixation. Consistent with the predictions of Eccentricity Bias Theory, place-selective regions are peripherally-biased²⁸. Thus, Eccentricity Bias Theory predicts that due to habitual fixation on faces, spatial computations in all face-selective regions, across both ventral and lateral processing streams, will be foveally-biased.*

An alternative hypothesis predicts that pRF properties and VFC will be different across face-selective regions of the ventral and lateral streams because these streams are optimized for different tasks with different computational demands. Face recognition requires fine spatial acuity afforded by central vision predicting that ventral face-selective regions, which are involved in face recognition, will be foveally-biased²⁸. However, social interactions often involve a group of people. As such, even when fixating on one face, processing social aspects of multiple faces in the group may require peripheral vision². Further, processing of dynamic information requires integrating optic flow across the visual field³⁰ and is faster in the periphery than in the fovea^{15,31}. Thus, the computational demands hypothesis predicts that pRFs and VFC in lateral face-selective regions, which are involved in social and dynamic processing of faces, will extend to the periphery.

3.) The “toonotopy” method seems clearly a better approach than the checkerboard methods for driving the ventral face areas. I would assume, however, that this reflects a combination response based on what something is (face/person), and where it is on the retina. So, it is perhaps not surprising that overall more high-level area voxels are driven by the cartoons. It would be

useful to know the extent to which the toons used in the pRF mapping show faces centered near the fixation point. That might account for at least some foveal bias in the more ventral areas—though it would not account for the more even distribution of pRF centers in the lateral areas. The reviewer asks if the faces in our mapping stimuli were more concentrated in the center of the visual field which may account for some of the foveal bias. We thank the reviewer for this comment. While we designed the experiment with a broad mapping of the visual field with all stimulus categories in mind, we agree that this is an important point. Therefore, we have added a new analysis of the distribution of face stimuli across the visual field. To do so, we created binary masks for each frame presented in the experiment, with white marking the location and extent of faces within the stimulus and black marking a non-face (other cartoon category or background). We then averaged these masks across all frames of the experiment to illustrate the distribution of faces across the visual field (new supplemental Figure S2, copied below). Results show that while the proportion of faces is highest in the center of the display and decreases as we approach the far periphery, there are still a substantial number of frames including face stimuli presented all the way out to 40° of eccentricity. Additionally, we note that: (i) using the same stimuli we find a peripheral bias in face-selective regions of the lateral stream, and (ii) prior studies reported foveal bias in ventral face-selective regions when contrasting responses to identical stimuli in the center and periphery of the visual field (e.g. Levy 2001, Weiner 2014; Kay 2015).

Figure S1: Location of face (vs. non-face) stimuli in the “toonotopy” experiment. To quantify the location of face stimuli across the visual field in our pRF mapping experiment, we created a binary mask for each frame of the experiment, where white indicated a face stimulus within the aperture and black indicated a non-face stimulus or background. Averaging these binary masks together, we see that face stimuli fairly comprehensively tile the circular aperture and are not solely located in the center of the display. To quantify this, we divided the image based on corresponding degrees of eccentricity when displayed to the subject and grouped pixels according to the four eccentricity bands used for analysis throughout the paper (0°–5°; 5°–10°; 10°–20°; 20°–40°). The proportion of frames containing face stimuli averaged across pixels in each eccentricity band were 0.0367, 0.0292, 0.0202, and 0.0128 for the 0°–5°, 5°–10°, 10°–20° and 20°–40° bands, respectively. This illustrates that while the proportion of faces is highest in the center of the display and decreases as we approach the far periphery, there are still a substantial number of frames including face stimuli presented out to 40°.

Action: We report the results on this new analysis on page 5 where we write: *“Faces in these cartoon images spanned the mapped visual field (Figure S1).”*

4.) I was surprised that mSTS responses were like V1 on this comparison (Fig 1 C), rather than the other face areas, including pSTS. (line 196). Any thoughts on why this might be? Given that the toons were static—is it possible that the form part of the stimulus did not drive voxels as much as it may have for the ventral face areas?

The reviewer asks about the comparison between the toon mapping stimuli and the checkerboard stimuli. The reviewer notes that the cartoon stimuli improve the model fits in all

face-selective regions except mSTS-faces and asks if this is due to using suboptimal static stimuli for this region.

The reviewer raises a good point. We designed these stimuli attempting to increase responses in all face-selective regions, including the lateral regions which are hypothesized to prefer moving stimuli. We did not include video clips in our mapping stimuli as we wanted to ensure that participants maintained fixation (which is critical for pRF mapping), and did not track motion in videos. To better drive the lateral stream regions, which prefer moving vs. stationary stimuli, we presented our stimuli at a higher rate, of 8 Hz, as other studies have shown that regions that prefer motion also prefer higher frequency stimuli (with peak sensitivity around 10Hz, Merigan & Maunsell 1993; Perrone & Theille 2001; Lui, Bourne & Rosa 2007). Our empirical data show that while this improved the number of voxels we could drive in pSTS-faces, it did not improve the number of mSTS-faces voxels we could drive with our mapping stimuli compared to checkerboards.

Action:

(i) We have added a justification of our choice of stimuli to the Methods. On pg. 17 we write: *“Due to our interest in higher-level category selective regions, bars of width 5.7° swept across high-contrast, colorful, cartoon stimuli that spanned a wide-field circular aperture ($40^\circ \times 40^\circ$ of visual angle). Images randomly changed at a rate of 8 Hz. The bars randomly revealed portion of the cartoon images and they largely include object parts rather than entire objects (Fig. 1A). We did not include video clips in our mapping stimuli as we wanted to ensure that participants maintain fixation (which is critical for pRF mapping) and do not track motion in videos. To better drive the lateral stream regions, which prefer moving vs. stationary stimuli^{3,13-17} we presented our stimuli at a higher rate of 8 Hz, as prior studies have shown that neurons in MT and other lateral stream areas that prefer motion also prefer higher frequency stimuli (with peak sensitivity around 10Hz)^{48,66}”.*

(ii) Additionally, we now acknowledge this difference in the effect of the stimuli in the results and speculate on its origin. On pg. 6 of the Results, we write: *“A 2-way repeated measures LMM ANOVA on the proportion of retinotopically-driven voxels with factors of experiment (toonotopy/checkerboards) and ROI (V1/IOG/pFus/mFus/pSTS/mSTS) revealed a significant ROI x experiment interaction in both the right ($F(5, 44)=6.3, p=.00017$) and left hemispheres ($F(5, 40)=6.0, p=.00032$). This interaction reflects the pronounced effect the type of mapping experiment had on driving responses in most face-selective regions, particularly within the ventral stream, but not V1, or mSTS-faces. For the latter, the combined change in stimuli (cartoons vs checkerboards) and presentation rate (8Hz vs. 2Hz) may have not been sufficient to drive neurons in the region, which prefer dynamic stimuli¹⁴ (Figures 1C, S2).”*

5.) Discussion – lines 468 -478 – I am not entirely convinced by “speed” account of why lateral areas have more peripherally extended input. It seems possible that any kind of coherent motion computation would require a larger area of the face (i.e., typical face motions would likely not “fit” in the fovea, but would instead involve the entire face and head). (Also, the

speed of expression processing in humans is likely also due in part to fast, simultaneous subcortical processing.)

We thank the reviewer for raising this point. We agree that speed is only one of several factors that may explain peripheral processing in the lateral stream.

Action: We have now expanded the Discussion to raise several possible functions of peripheral pRFs in the lateral stream.

In the Discussion, pg. 14, we now write: *“Our finding of pRFs extending to the periphery in lateral face-selective regions raises a new question: what is the benefit of these peripheral computations? We propose several non-mutually exclusive hypotheses that can be tested in future research. One possibility is that peripheral computations in lateral face-selective regions and their direct white matter connections to peripheral EVC may facilitate rapid processing of dynamic^{24,47} and transient¹⁵ face information, as the speed of visual processing is faster in the periphery than the fovea³¹. Another possibility is that large and peripheral pRFs in lateral face-selective regions may enable integrating information across the entire visual field, which is important for computing shape and motion information from optical flow³⁰ across the entire face or whole person⁴⁸. A third possibility is that in social situations involving a group of people, peripheral pRFs in lateral face-selective regions may allow for inferring social information from faces in the periphery, which are not directly fixated upon, such as an eye-roll or an encouraging smile.”*

6.) A final general comment is the use of the term “spatial processing” throughout, which I found confusing. Arguably, lateral (dorsal stream) areas contribute far more to spatial orienting, etc. and may require a more complete representation of ego-centric space than ventral regions. This might account for more complete EVC space representations. The term spatial processing for lateral areas makes sense. It is not clear what “spatial processing” means in ventral areas. Arguably, in natural viewing, foveation is used to examine face features (eyes, mouth, etc.) in more detail. It is possible that the advantage of foveal input to ventral areas is that it may be the most efficient place to find information useful for categorization. It does not necessarily follow that the function of these ventral areas is to better represent that part of visual space. This seems to be way the term is used in the paper. This may be splitting hairs, but I think that the term, as applied here to ventral and lateral areas, is not meant in entirely the same way. The reviewer raises an important point, asking: what do we mean by spatial processing in the context of the ventral stream? In brief, classic theories of visual streams suggest that processing in ventral temporal cortex (VTC) is detached from the spatial location of the stimuli – often using the term “position invariant” to describe the neural characteristics of VTC. Our study adds important evidence showing that this classic framework is not supported by empirical data, and that in fact, processing in VTC is not position invariant, but actually depends on the location of the stimulus in the visual field.

Action: We have replaced the term “spatial processing” to “spatial computations” throughout the manuscript and define it on pg. 2 of the introduction, where we write: *“A basic characteristic of the visual system is the spatial computation by the receptive field²² (RF), which is akin to a filter that processes visual information in a restricted part of visual space.”*

We also justify why we sought to examine pRFs in high-level visual cortex, on pg. 2 we write: *“While classic theories have hypothesized that RFs are mainly a characteristic of early and intermediate visual areas, accumulating evidence suggests that pRFs are also a characteristic of high-level visual areas²⁰⁻²³.”*

Reviewer #2 (Remarks to the Author):

This work, led by Dawn Finzi of the the Grill-Spector lab, documents some interesting differences between lateral and ventral face-selective regions of the human brain. The principal finding is that face-selective regions in the superior temporal sulcus (lateral stream) are biased toward more peripheral regions of the visual field, with coarser spatial sampling (larger pRFs), compared to face areas on the ventral stream. This result is demonstrated clearly in a series of analyses, and complemented by the interesting anatomical finding that fiber tracts, derived from diffusion weighted imaging, preferentially connect foveal regions of V1, V2, and V3 to ventral face areas (matching the measured spatial selectivity) whereas more peripheral regions of V1, V2, and V3 are preferentially connected to lateral face areas. The work nicely brings together findings from three important areas in human visual neuroscience; category-selectivity in cortex, retinotopy, and anatomical pathways. It may serve as a useful benchmark for many other studies. For example, by describing systematic differences between lateral and ventral face area properties, the authors will help constrain questions of homology between human face areas and those of other species. This work will likely also lead to interesting followups on how the lateral and the ventral face areas communicate with one another during tasks requiring the functions of both streams. In short, the clear set of findings on an important topic make this an impactful paper. I find most of the analyses clear and convincing, and as such I have only minor concerns.

We thank the reviewer for their enthusiastic summary of our study and its potential impact.

Minor

1. Eccentricity analyses (e.g., figure 2B) tend to go out to 40 deg. The mapping stimulus only went out to 20 deg. While it is possible to effectively map a pRF whose center is outside the maximal stimulus extent, the precision of the eccentricity estimates drops. It might be more appropriate to bin data beyond some threshold (say 20 deg, or maybe 25 deg) rather than to assume the eccentricity can be measured accurately all the way out to 40 deg. Another possibility would be to limit voxels to those whose center is within some criterion of the stimulus extent dependent on pRF size (say, within 1 STD). At a minimum, the authors should consider whether uncertainty at these high eccentricities could substantially affect any key results or conclusions. (I would think probably not!)

We agree with the reviewer that the accuracy of pRF estimates is higher within the range of the mapping stimulus (here 20° from fixation). The reviewer suggests that we repeat our analyses related to Figure 2B within this range to ensure that results are not due to reduced accuracy in the far periphery (20°-40°). We thank the reviewer for this suggestion.

Action: In order to ensure that our results are not affected by any decrease in the accuracy of our pRF estimation outside of the maximal stimulus extent, we reanalyzed the data limiting eccentricities to the central 20°. These analyses are presented in new Supplemental Figure S4 which shows the results of the same analyses presented in Figures 2B and 4A but only restricted to pRFs with centers within the central 20°.

Results of these new control analyses, shown in new Supplemental Figure S4, replicate the main results for the central 20°. Analyses related to Supplemental Figure S4 show a significant eccentricity band x stream interactions in both hemispheres (right: $F(2, 321) = 111.4, p < 2.2 \times 10^{-16}$; left: $F(2, 276) = 58.9, p < 2.2 \times 10^{-16}$) with post-hoc Tukey's tests showing a significantly higher proportion of centers in the 0–5° eccentricity band in ventral vs. lateral face-selective regions (right: $t(321)=10.7, p<.0001$; left $t(276)=6.0, p<.0001$) and a significantly lower proportion of centers for ventral vs. lateral regions in the now most peripheral 10–20° band (right: $t(321)=10.4, p<.0001$; left $t(276)=8.7, p<.0001$). These results are reported in the caption of new Supplemental Figure S4 and referenced in pg. 7 of the manuscript, where we write: *"...as well as a significantly lower proportion of centers for ventral vs. lateral regions in the two most peripheral eccentricity bands. These differences persist even if analyses are limited to the maximal extent of the stimulus (central 20°, Figure S4)."*

We had also included a visualization of the pRF centers limited to the central 10° for each ROI, in order to more clearly display the differences in density of pRF centers within the central portion of the visual field (Supplementary Figure S3), referenced on pg. 7 of the manuscript, where we write: *"Visualizing pRF centers across participants for each ROI (Figure 2A) reveals differences across ventral and lateral regions. pRF centers of ventral face-selective regions, IOG, pFus and mFus, are largely confined within the central 10° (Figure 2, S3)."*

These plots enhance the visualization of the data with the central 10° and better visualize the distribution of pRF of the first 2 bars for each region in Figure 3B.

Figure S3: Distribution of pRF centers for the central 10 degrees
 Each dot is a pRF center. Data are shown for all pRFs across all participants for both hemispheres. Dark colors: Left hemisphere. Corresponding main text figure: figure 2A.

Figure S4: Control analysis limiting model fits to 20° of eccentricity. (A) Proportion of pRF centers of face-selective ROIs and CoS-places across eccentricities bands when model fits were constrained to the central 20°. For each participant and ROI, the proportion of pRF centers in each of three eccentricity bands (0°–5°; 5°–10°; 10°–20°) was calculated. As in the main results (figure 2B), a 2-way repeated-measures ANOVA on the proportion of centers for each hemisphere separately with factors of eccentricity band (0–5°/5–10°/10–20°/20–40°) and stream (ventral: IOG/pFus/mFus and lateral: pSTS/mSTS) confirms significant eccentricity band x stream interactions in both hemispheres (right: $F(2, 321) = 111.4, p < 2.2 \times 10^{-16}$; left: $F(2, 276) = 58.9, p < 2.2 \times 10^{-16}$). Bars: mean across participants; Each dot is a participant; Error bars: \pm SE. Corresponding main text figure: figure 2B. (B) Average visual field coverage of the central 20° for face-selective and CoS-places ROIs across participants. Corresponding main text figure 4A.

2. Density vs eccentricity. The density vs eccentricity functions of different areas comprise a key result (e.g., Figure 4b). The plots themselves are pretty convincing, but summarizing the distributions by linear fits is probably not optimal. The data look like they might be better described by a sigmoid of some kind, or perhaps piecewise linear (limited to a small number of pieces). Of course a line has the advantage of interpretation - one can summarize a single parameter (slope). Nonetheless, it looks a line is a poor fit to some of the distributions, esp the ventral face areas in Fig 4B.

The reviewer suggests that sigmoid line fits would be better function to fit the data in Figure 4B. Thank you for suggesting this complementary analysis. We were initially drawn to a linear function due to the simplicity and interpretability of the parameters, but we agree that the ventral face-selective regions are better approximated by a more complex function.

Action: Based on the reviewer’s suggestion to try fitting a sigmoid of some kind to the data, we added another analysis in which we fit a generalized logistic function to the data and then evaluated differences in the parameters of the logistic function across the ventral and lateral

face-selective regions. Results of this additional analysis based on the generalized logistic fit replicate the results of the linear fit.

We've added the following sections to the Methods and Results to reflect the inclusion of this analysis:

Pg. 20 (Methods): *"To assess differences in VFC across ROIs, we fitted both linear and generalized logistic functions relating pRF density to eccentricity in each ROI and participant, and then compared the parameters (across participants) of different ROIs in each stream. The linear function was of the form $y = ax + b$, where a represents the slope and b the y-intercept. The generalized logistic function was of the form: $y = a + \frac{b-a}{1+10^{(c-x)*d}}$ where fitted parameters a , b , c , and d represented the lower asymptote, the upper asymptote, the inflection point, and the steepness of the curve, respectively. Mean adjusted R^2 was high for the linear model (mean \pm SE adjusted R^2 across ROIs and subjects: right = $.81 \pm .01$, left = $.80 \pm .02$) and was, unsurprisingly given the additional number of parameters, even higher for the generalized logistic function (mean \pm SE adjusted R^2 across ROIs and subjects: right = $.97 \pm .01$, left = $.96 \pm .02$)".*

Pg. 9-10: *"We summarized the relationship between pRF density and eccentricity by fitting both linear and generalized logistic functions for each participant's pRF density curve per ROI as the ventral regions appear to be best approximated by a sigmoidal curve, while the lateral face-selective regions appear to be best approximated by a linear fit. Irrespective of the model fitting approach, we find significant differences in the fitted parameters between ventral and lateral face-selective regions."*

Pg. 10: *"Similarly, the parameters for both the inflection point and the lower asymptote in the generalized logistic function are significantly different between ventral and lateral regions in both hemispheres, such that ventral face-selective regions have smaller valued lower asymptotes (paired t-tests; right: $t(24)=-4.6$, $p=0.00011$; left: $t(22)=-4.2$, $p=.00041$) and inflection points (paired t-tests; right: $t(24)=-4.0$, $p=0.00055$; left: $t(22)=-2.3$, $p=0.034$) than lateral face-selective regions."*

3. The authors' estimates of pRF centers go out to 40 deg, and they find a fairly large number of such voxels in lateral face areas. How do the authors reconcile this with the fact that, as they note in several places in the text, faces are typically foveated? A face would need to be unusually close for it to extend out to 40 deg when foveated. Perhaps the point is that even though faces are *often* foveated, they are not *always* foveated? Or perhaps these areas play a role in choosing face targets in the periphery to foveate? In any case, it would help to comment on this issue. Simply indicating that these areas process dynamic information and that dynamic information is represented differently in the periphery does not by itself explain why it would be useful to have peripheral face representations.

The reviewer asks how we reconcile that pRFs of lateral face-selective regions extend to 40 degrees from the center when faces are typically foveated upon and need to be unusually close to extend out to 40 degrees. Thank you for suggesting this important discussion point.

Action: We have elaborated in both the Introduction and Discussion why tasks related to the lateral stream may involve peripheral vision even if one is fixating on face.

(i) In the Introduction, pg.3, we write: *“However, social interactions often involve a group of people. As such, even when fixating on one face, processing social aspects of multiple faces in the group may require peripheral vision². Further, processing of dynamic information requires integrating optic flow across the visual field³⁰ and is faster in the periphery than in the fovea^{15,31}. Thus, the computational demands hypothesis predicts that pRFs and VFC in lateral face-selective regions, which are involved in social and dynamic processing of faces, will extend to the periphery.”*

(ii) In the Discussion, pg. 14, we write: *“Our finding of pRFs extending to the periphery in lateral face-selective regions raises a new question: what is the benefit of these peripheral computations? We propose several non-mutually exclusive hypotheses that can be tested in future research. One possibility is that peripheral computations in lateral face-selective regions and their direct white matter connections to peripheral EVC may facilitate rapid processing of dynamic^{24,47} and transient¹⁵ face information, as the speed of visual processing is faster in the periphery than the fovea³¹. Another possibility is that large and peripheral pRFs in lateral face-selective regions may enable integrating information across the entire visual field, which is important for computing shape and motion information from optical flow³⁰ across the entire face or whole person⁴⁸. A third possibility is that in social situations involving a group of people, peripheral pRFs in lateral face-selective regions may allow for inferring social information from faces in the periphery, which are not directly fixated upon, such as an eye-roll or an encouraging smile.”*

Very minor.

1. The authors often refer to the ventral stream as being involved in perception of static properties. I think this language can be improved and clarified. While faces may be static during fMRI experiments, they are presumably rarely entirely static in natural viewing. Perhaps the authors mean “stable” properties such as identity that do not change over time. The point is, while a property might be stable, a face typically is not (entirely) static. Static vs dynamic face perception sounds more like a description of an experimental paradigm than the functional selectivity of a brain area.

We have used these terms as they are prevalent in the literature. To clarify what we mean we modified the sentence on pg. 13 to: *“A substantial body of research³⁻⁵ highlights differences in the computational goals of ventral and lateral streams: ventral face-selective regions are thought to be involved in face recognition, while lateral regions are thought to be involved in dynamic^{4,12,13} and social aspects of face perception^{8,9,18,46}.”*

2. On p 2 (line 47) the authors state that spatial processing is the basic computation of the visual system. Perhaps. Others might argue that color, depth, motion, etc are equally “basic”. Perhaps change to “A basic” ...?

We thank the reviewer for pointing this out as well and for the suggestion, which we have implemented. We have changed the sentence on pg. 2 as follows: *“A basic characteristic of the*

visual system is the spatial computation by the receptive field²² (RF), which is akin to a filter that processes visual information in a restricted part of visual space.”

3. Throughout the text, the authors refer to EVC, which they define as V1, V2, and V3. It is true that many authors sometimes do this. Nonetheless, a few sentences of justification explaining why these areas are similar to each other and why they are different from other areas would help motivate the grouping. This might come from cluster analysis in published work, or just evidence that these areas have the clearest retinotopic maps, or that they show relatively similar response patterns, etc.

Thank you for pointing out this oversight. The reviewer asks for a justification of why we combine V1, V2, and V3 to an early visual cortex (EVC) ROI.

Action: We have now added a justification for our decision to create an EVC ROI within the Methods under the section: *Definition of V1-V3 (EVC)*. On pg. 18, we write: *“We chose this particular grouping for two main reasons: (i) V1, V2, and V3 share an eccentricity representation and are often grouped together in the hierarchy of the visual stream, separate from hV4 which that is classified as an intermediate area, and (ii) We were concerned that seeding for tractography may be too narrow if limited to one retinotopic area. For these reasons, we generated an EVC ROI that is the union of V1, V2, V3 to ensure that we had a full and accurate representation of the distribution of streamlines from eccentricity bands in early visual cortex.”*

4. P. 115 the authors refer to standard retinotopy experiments as containing flickering checkerboards. Perhaps some do, but the most common ones I know of (eg Dumoulin and Wandell, 2008) tend to have drifting contrast patterns rather than flickering. Moreover the patterns are not always checkerboards, but often dartboards or other achromatic contrast patterns (such as Kay et al 2013, JNP). Hence “contrast patterns” would probably be more appropriate than “checkerboards”.

We thank the reviewer for this comment and have replaced the mention of “checkerboards” on that line with “high contrast patterns”.

5. Novelty of toonotopy. It is fine for the authors to describe the toonotopy as a novel design, since this specific design is novel. Nonetheless, there have been many retotopy studies testing patterns other than simple contrast patterns, including Point-Light Walkers (Saygin, Sereno); words (Le ... Wandell, 2017), the HCP retinotopy experiments (Benson et al 2018); natural movies like Buffy the Vampire Slayer (Sereno, though perhaps only in abstract form). In most of these cases, the stimulus was chosen for similar reasons to this paper, namely to get better activation in higher level visual areas. I think it would be appropriate to cite some of these, perhaps whichever came earliest.

Thank you for highlighting these relevant contributions. To credit previous research, we have changed the description of toonotopy experiment referencing Le et al, 2017 and the HCP retinotopy experiments (Benson et al 2018). On pg 3. in the introduction, we now write: *“To test these hypotheses, we designed a novel pRF mapping experiment optimized to map pRFs in high-level visual regions (Figure 1A inspired by other experiments using complex stimuli including objects^{32,33}, faces²¹, and words²².”*

6. Line 361 is missing a reference. It literally says “{ref}”.

We very much appreciate the reviewer for noticing this oversight. We have corrected the mistake to include the appropriate reference to Pestilli et al., 2014.

Signed,
Jon Winawer
NYU

Reviewer #3 (Remarks to the Author):

Finzi and colleagues examined the pRF and white matter connections in the ventral and lateral face-selective areas. They examined the hypothesis that pRF and connections from EVC are foveal in the ventral stream (consistent with previous reports) whereas the lateral face areas represent more peripheral visual field information related to their role in processing dynamic facial information.

The paper is very well written and the analyses are well performed and presented. Based on the title and the abstract, though, I was expecting to see much clearer findings than those that are actually reported. The results of the ventral face areas are foveally biased as expected and was found previously, but the lateral areas do not show as consistent picture as described. The pSTS and mSTS are often showing different patterns of results. Also, the mSTS is found in a very small number of participants (6 out of 21) and it is therefore hard to draw conclusions on the entire lateral face-stream based on such small sample size. It also makes the statistical analysis more problematic when an interaction with ROI is examined for such a small number of subjects in one of the ROIs relative to the others.

This could have been resolved if the regions would have been localized with a dynamic face localizer. The STS face areas – in particular the more anterior ones that were hardly found in the current study - show twice as much response to dynamic than static faces, and the size of the regions responding to dynamic face stimuli in the STS is much larger. However, face-selective areas were defined with static images of faces and non-face stimuli, which generated face-selective activations in the ventral face areas as expected, whereas activations of the lateral face areas were less consistent, in particular, in the mSTS. Also the pSTS, that is typically found also for static faces, is expected to be much larger for dynamic faces than the one that is found for static faces.

Given that the goal of the study was to assess differences between lateral and ventral face regions, and in particular differences that may derive from the different roles that the two streams play in the processing of dynamic and static face stimuli – the regions that were defined may not be suitable for testing these hypotheses. Indeed, results in the mSTS are not always consistent with the hypothesis and the pSTS and mSTS sometimes generate different patterns of responses (e.g. Figure 3; Figure 7).

This is very unfortunate because the authors do have the capability and expertise to perform this work well. However, based on the current data, conclusions on the different computations of the two face streams based on foveal vs peripheral visual processing is not well supported as could have been.

We thank the reviewer these comments. The reviewer raises several important points, which we address in turn.

First, the reviewer is concerned that differences between the lateral and ventral stream face-selective regions are based on a minority of subjects as one of the lateral stream regions, mSTS-faces, was identified in only 6 out of 21 participants.

Second, the reviewer suggests that our localizer, which is based on static images, is suboptimal for localizing face-selective regions in the lateral stream. They believe that if we had used a dynamic localizer instead, we would have found the mSTS patches in more participants, and those patches would also be larger.

Third, the reviewer suggests that these factors (low number of participants and small ROIs) may have contributed to spurious differences between pSTS-faces and mSTS-faces, which are both lateral stream face-selective regions.

We wish to clarify that the low N for mSTS-faces was not due to our inability to localize the mSTS from the localizer as the reviewer suggests, as we were able to localize right mSTS-faces with our static localizer in 18 of the original 21 participants. The reason that we report pRF data for only 6 out of 21 participants for this ROI is that we had two criteria for subject inclusion in the pRF analyses: (1) identification of the face-selective region using the localizer and (2) that the pRF model explained at least 10% of variance in at least 25 voxels in the ROI. We used this second criteria to ensure that we have high quality data from each ROI and that we have a sufficient number of voxels within a ROI to generate a coverage map, as we do all our analysis in a subject-by-subject basis.

Given that the second exclusion criterion, which is based on the main experiment – the toonotopy pRF mapping study, was the main data bottleneck, we believe that the reviewer's suggestion to scan subjects with a dynamic localizer would not ameliorate the low N for the mSTS-faces in pRF mapping study. Additionally, COVID restrictions have prevented us from being able to rescan these participants with a dynamic localizer.

Action: We thank the reviewer for suggesting that including more data for mSTS-faces and extending the ROI would strengthen the results of this paper. Therefore, we implemented the following steps to increase the number of subjects in our study:

(1) Prior to the COVID pandemic, seven additional participants were scanned with the original localizer and the toonotopy pRF mapping experiment for the purposes of another study just before the scanner shut down in March 2020. We have added new data from these additional subjects to the manuscript. This increased the number of subjects in all ROIs and particularly in mSTS-faces which is now identified in 24 out of 28 subjects in the right hemisphere and 12 in

the left hemisphere, strengthening the paper overall. In pg. 18, we detail in how many subjects we identified each ROI: *inferior occipital gyrus (IOG-faces, right hemisphere: N=27; left hemisphere: N=24)*, *posterior fusiform gyrus (pFus-faces, right: N=25; left: N=23)*, *mid fusiform gyrus (mFus-faces; right: N=23; left: N=25)*, *posterior superior temporal sulcus (pSTS-faces, right: N=25; left: N=23)*, and *medial superior temporal sulcus (mSTS-faces, right: N=24; left: N=12)*. These additional number of subjects strengthened our confidence in our findings and allowed us to update our threshold to a more stringent 20% variance explained per voxel, though we have replicated all results with the original threshold (see **Supplementary Appendix**).

(2) To address the issue that the mSTS-faces ROI that we identified was smaller than expected from a dynamic localizer, we assessed the mean size of this region from prior studies using a dynamic localizer. In Sliwinska et al., 2020, mSTS-faces was defined from a dynamic localizer using a 5mm sphere centered around the peak voxel of activation. We sought to match our mSTS-faces ROI, which we localized in the 24 out of 28 participants, to their ROI size. This was implemented by placing a 1 cm disk ROI on the center of each of our functionally defined mSTS-faces in each participant. This involved three changes from the Sliwinska 2020 procedure: 1) We chose to use the center instead of the peak voxel of activation as prior data from our lab (Weiner 2010) investigating the reproducibility of face-selective regions across sessions and experiments showed that the center of the ROI is a stable attribute of the ROI. 2) We used a disk ROI instead of a sphere ROI in order to restrict the ROI to the gray matter as neurons reside only in the gray matter. 3) We used a 1 cm instead of 5 mm radius to match the surface area of the disk ROI to surface area of the sphere ROI, which is approximately 314mm². We updated the Methods on pg. 19 to reflect this definition of the ROI. Consistent with the reviewer's prediction, increasing the size of the ROI enabled us to capture more voxels that were significantly modulated during the toonotopy experiment. After applying inclusion criteria from the pRF mapping experiment (including ROIs which had at least 10 voxels in which the pRF model explained at least 20% of their variance), we now present data from 13 subjects for right mSTS-faces (compared to 6 before) and 7 subjects for the left mSTS-faces (compared to 2 before).

Results of these new analyses are incorporated in all figures showing mSTS data. A direct comparison of key results (Figures 2, 4, and 7) between the functionally defined mSTS ROI and the 1 cm disk ROI is included in new Supplemental Figure S8, copied below).

pSTS-faces is defined using the same functional criteria as all other ROIs in this study and remains unchanged from the original manuscript. We note that the size of pSTS-faces is already substantial in both hemispheres (surface area, right pSTS: mean = 234 mm², left pSTS: mean = 171mm²) and prior research indicates that the gain by dynamic stimuli is modest in this region.

New Figures 2, 3, and 4 now show data from all 28 subjects who participated in the toonotopy experiment, and contain data from bilateral mSTS with a 1 cm disk ROI centered on the functionally-defined face ROI from the localizer experiment. All relevant statistics have been updated as well. This added subjects in two ways: 1) we have additional participants for all ROIs and particularly right mSTS-faces as noted above and 2) we now have a sufficient number of participants to include left mSTS-faces in the pRF mapping experiment, replicating in the left mSTS-faces, our findings with right mSTS-faces.

With the addition of subjects and use of the disk ROI, the results of the original manuscript have been replicated and strengthened. We find significant differences in the distribution of pRFs, visual field coverage, and fiber endpoint density between ventral (IOG, pFus, mFus) and lateral face-selective regions (pSTS, mSTS).

The key pRF findings in the Results are:

Main text pg. 7: "A 2-way repeated measures LMM ANOVA on the proportion of centers with eccentricity band (0–5°/5–10°/10–20°/20–40°) and stream (ventral: IOG/pFus/mFus and lateral: pSTS/mSTS) as factors revealed a significant eccentricity band x stream interaction in both hemispheres (right: $F(3, 428)=77.1$, $p<2.2\times 10^{-16}$; left: $F(3,376)=41.0$, $p=2.2\times 10^{-16}$). Post-hoc Tukey's tests establish that this is driven by a significantly higher proportion of centers in the most foveal 0–5° eccentricity band in ventral vs. lateral face-selective regions (proportion higher in ventral than lateral 0–5°: right: 0.54 ± 0.04 , $t(428)=12.2$, $p<.0001$; left: 0.41 ± 0.06 , $t(376)=7.1$, $p<.0001$), as well as a significantly lower proportion of centers for ventral vs. lateral regions in the two most peripheral eccentricity bands (proportion lower in ventral than lateral, 10–20° right: 0.32 ± 0.04 , $t(428)=-7.4$, $p<.0001$; left: 0.30 ± 0.06 , $t(376)=-5.2$, $p<.0001$; 20–40° right: 0.23 ± 0.04 , $t(428)=-5.2$, $p<.0001$; left: 0.32 ± 0.06 , $t(376)=-5.54$ $p<.0001$)."

*Main text pg. 8: "Results show that in both hemispheres pRFs were significantly larger in lateral than ventral face-selective regions (paired t-tests; right: $t(24)=-4.3$, $p=.00025$; left: $t(22)=-6.1$, $p=3.6\times 10^{-6}$). Differences between ROIs were significant (1-way repeated measures LMM ANOVAs on median pRF size, right ROIs: IOG/pFus/mFus/pSTS/mSTS, $F(4,86)=13.6$, $p<1.3\times 10^{-8}$; left ROIs: IOG/pFus/mFus/pSTS/mSTS, $F(4,74)=22.0$, $p=5.4\times 10^{-12}$), and were driven by significant differences between both pSTS-faces and each of the ventral face-selective regions (post-hoc Tukey tests, all $t_s>4.8$, $p_s\leq .0001$, **Table S1**) and mSTS-faces and each of the ventral face-selective regions (post-hoc Tukey tests, all $t_s>3.5$, $p_s\leq .0064$, **Table S1**)."*

Main text pg. 10: “Results reveal (i) significant differences between the average slopes of ventral and lateral face-selective regions (paired t-tests; right: $t(24)=-9.8$, $p=7.4 \times 10^{-10}$; left: $t(22)=-6.9$, $p=6.1 \times 10^{-7}$), whereby slopes for ventral face-selective ROIs were more negative than for lateral face-selective ROIs and (ii) significant differences between the average slopes of individual face-selective ROIs (right: $F(4,85)=47.6$, $p<2.2 \times 10^{-16}$; left: $F(4,90)=23.9$, $p=1.7 \times 10^{-13}$, 1-way repeated measures LMM ANOVAs on the slopes with factor ROI). Specifically, slopes in lateral face-selective regions—pSTS-faces and mSTS-faces—were significantly closer to zero than any of the ventral face-selective regions (all $t_s>4.1$, $p_s \leq .0008$, post-hoc Tukey tests, **Table S3**). Additionally, bilateral pFus-faces and right mFus-faces had significantly more negative slopes than IOG-faces (all $t_s<-3.3$, $p_s \leq .012$, **Table S3**), indicating that the former ROIs have a larger foveal bias than the latter. Similarly, the parameters for both the inflection point and the lower asymptote in the generalized logistic function are significantly different between ventral and lateral regions in both hemispheres, such that ventral face-selective regions have smaller valued lower asymptotes (paired t-tests; right: $t(24)=-4.6$, $p=0.00011$; left: $t(22)=-4.2$, $p=.00041$) and inflection points (paired t-tests; right: $t(24)=-4.0$, $p=0.00055$; left: $t(22)=-2.3$, $p=0.034$) than lateral face-selective regions.”

As another test we also repeated these analyses using a more lenient threshold of 10% variance explained per voxel, which allowed us to include more data but may include more poorly fit, noise voxels. All key results replicate (see Supplementary Appendix).

(3) We greatly thank the reviewer for these comments as we believe they have led to significant improvement of the manuscript. While the reviewer is correct that the second hypothesis (computational demands hypothesis) predicts that pRF and VFC properties should be more similar within than across streams, we would like to underscore that both hypotheses do not preclude changes in pRF and VFC properties across ROIs within a processing stream. For example, we have shown previously that as one ascends the ventral stream from IOG to pFus to mFus-faces there are systematic increases in pRF size and ipsilateral coverage (Kay 2015).

Nonetheless, with the addition of participants and the change to more similar sized pSTS- and mSTS-faces, it is evident that the lateral face ROIs are more similar to each other than to the ventral ROIs, as detailed below:

With respect to the location of pRFs: pSTS-faces and mSTS-faces show a similar distribution of pRF centers bilaterally, with centers distributed across eccentricity bands, including more than 50% in the two most peripheral eccentricity bands, but ventral ROIs show a pronounced foveal bias (Figure 2).

With respect to pRF size, both pSTS- and mSTS-faces have larger pRFs than the ventral face-selective regions (Figure 3A, mean pRF size: right IOG-faces = 7.1, right pFus-faces = 9.7, right mFus-faces = 9.1, right pSTS-faces = 21.2, right mSTS-faces = 19.4). However, we also find that pSTS-faces has the largest receptive fields, with sizes that are relatively unchanged across eccentricity bands (Figure 3B).

With respect to visual field coverage, both pSTS-faces and mSTS-faces have pRFs that extend to the far periphery, contrasting the ventral ROIs that have a foveal bias (Figure 4). Nonetheless, mSTS-faces has more diffuse visual field coverage than pSTS-faces as it has more ipsilateral pRFs (Figure 2A), mirroring the pattern of more ipsilateral pRFs in mFus-faces than pFus-faces within the ventral stream hierarchy (Kay 2015).

(4) While we were unable to scan additional subjects using dMRI for the second experiment, we had previously localized right mSTS-faces in 14 of the 16 subjects who participated in dMRI and as we had no additional inclusion criteria for this experiment (different than the pRF mapping experiment) we already had a sufficient number of participants for all ROIs except left mSTS-faces: number of participants for each ROI: IOG-faces, right hemisphere: $N=15$; left hemisphere: $N=13$; pFus-faces, right: $N=13$; left: $N=13$; mFus-faces, right: $N=13$; left: $N=14$; pSTS-faces, right: $N=14$; left: $N=12$, and mSTS-faces, right: $N=14$.

To ensure that reporting is consistent across the manuscript, we used the same 1 cm disk ROI placed on the center of the right mSTS-faces ROI for the dMRI data analyses and updated the results accordingly. Figures 6 & 7 now contain mSTS-faces data based on the 1 cm disk ROI.

Figure 7 summarizes the diffusion findings, showing that right pSTS-faces and right mSTS-faces have a similar distribution of endpoints across eccentricity bands, which is more uniformly distributed across eccentricity bands compared to ventral face-selective regions.

Supplementary Appendix: Validating key results with a more lenient threshold of 10% variance explained (voxel level) by pRF model

As a further test of our results, we have reexamined the data with a more lenient threshold of 10% variance explained by pRF model per voxel. This allows inclusion of more data as we include ROIs with at least 10 voxels which exceed this variance explained threshold. All the key effects replicate with this threshold.

Difference in proportion centers across streams (related to pg. 7): A 2-way repeated measures LMM ANOVA on the proportion of centers with eccentricity band (0–5°/5–10°/10–20°/20–40°) and stream (ventral: IOG/pFus/mFus and lateral: pSTS/mSTS) as factors revealed a significant eccentricity band x stream interaction in both hemispheres (right: $F(3, 460)=101.0, p<2.2\times 10^{-16}$; left: $F(3,384)=56.7, p=2.2\times 10^{-16}$). Post-hoc Tukey's tests establish that this is driven by a significantly higher proportion of centers in the most foveal 0–5° eccentricity band in ventral vs. lateral face-selective regions (proportion higher in ventral than lateral 0–5°: right: $0.56\pm 0.04, t(460)=13.9, p<.0001$; left: $0.43\pm 0.05, t(384)=8.5, p<.0001$), as well as a significantly lower proportion of centers for ventral vs. lateral regions in the two most peripheral eccentricity bands (proportion lower in ventral than lateral, 10–20° right: $0.27\pm 0.04, t(460)=-6.8, p<.0001$; left: $0.28\pm 0.05, t(384)=-5.6, p<.0001$; 20–40° right: $0.32\pm 0.04, t(460)=-7.9, p<.0001$; left: $0.36\pm 0.05, t(384)=-7.1, p<.0001$).

Differences in pRF size across streams (related to pg. 8): Results show that in both hemispheres pRFs were significantly larger in lateral than ventral face-selective regions (paired t-tests; right: $t(26)=-3.9, p=.00057$; left: $t(23)=-5.5, p=1.2\times 10^{-5}$). Differences between ROIs were significant (1-way repeated measures LMM ANOVAs on median pRF size, right ROIs: IOG/pFus/mFus/pSTS/mSTS, $F(4,93)=23.2, p<2.6\times 10^{-13}$; left ROIs: IOG/pFus/mFus/pSTS/mSTS, $F(4,76)=15.6, p=2.4\times 10^{-9}$), and were driven by significant differences between both pSTS-faces and each of the ventral face-selective regions (post-hoc Tukey tests, all $t_s>5.0, p_s<.0001$).

Difference in visual field coverage across streams (related to pg. 10): Results reveal (i) significant differences between the average slopes of ventral and lateral face-selective regions (paired t-tests; right: $t(25)=-10.5, p=1.1\times 10^{-10}$; left: $t(25)=-10.2, p=2.4\times 10^{-10}$), whereby slopes for ventral face-selective ROIs were more negative than for lateral face-selective ROIs and (ii) significant differences between the average slopes of individual face-selective ROIs (right: $F(4,92)=53.0, p<2.2\times 10^{-16}$; left: $F(4,85)=34.3, p<2.2\times 10^{-13}$, 1-way repeated measures LMM ANOVAs on the slopes with factor ROI). Specifically, slopes in lateral face-selective regions—pSTS-faces and mSTS-faces—were significantly closer to zero than any of the ventral face-selective regions (all $t_s>4.7, p_s<=.0001$, post-hoc Tukey tests). Additionally, bilateral pFus-faces and right mFus-faces had significantly more negative slopes than IOG-faces (all $t_s<-3.3, p_s<=.0098$), indicating that the former ROIs have a larger foveal bias than the latter. Similarly, the parameters for both the inflection point and the lower asymptote in the generalized logistic function are significantly different between ventral and lateral regions in both hemispheres, such that ventral face-selective regions have smaller valued lower asymptotes (paired t-tests;

right: $t(25)=-3.6, p=0.0015$; left: $t(23)=-4.9, p=5.4 \times 10^{-5}$) and inflection points (paired t-tests; right: $t(25)=-3.2, p=0.0030$; left: $t(23)=-3.6, p=0.0015$) than lateral face-selective regions.

Reviewer #1 (Remarks to the Author):

The authors have addressed my concerns from the previous version of the manuscript and I am happy to recommend publication.

Reviewer #2 (Remarks to the Author):

The revised manuscript reflects effort and sensitivity to the initial reviews, both by me and by the other reviewers. The additions to the manuscript, such as the sigmoid fits in figure 4 and supplemental analysis with restricted eccentricities, strengthen the paper. The motivation and discussion are now clearer as well.

I have only remaining suggestion (which should not require re-review by me). In response to my point 3 (about face perception at 40°), there is one other point the authors might make (if they agree with it). Specifically, the difference in retinotopy for the different face areas might predict that social judgments from face stimuli will decline less in performance as a function of eccentricity than identity judgments. It is nice to make behavioral predictions from novel neural measures where possible. In any case, just a thought.

Overall, interesting and compelling paper! Congratulations.

Jonathan Winawer

Reviewer #3 (Remarks to the Author):

The authors addressed my main concern. The revised manuscript includes data from additional participants in particular for mSTS. Results are now more convincing with respect to the differences in pRF between the ventral and lateral face areas.

I have only a couple remaining comments:

1. Please add effect size and confidence intervals to all the reports of statistical effects.
2. I am wondering to what extent these findings are specific to face areas or are a general difference between ventral areas and the STS.

Point by point answers to reviewer comments are indicated in blue below each comment.

Reviewer #1 (Remarks to the Author):

The authors have addressed my concerns from the previous version of the manuscript and I am happy to recommend publication.

We thank the reviewer for their helpful comments and their positive recommendation.

Reviewer #2 (Remarks to the Author):

The revised manuscript reflects effort and sensitivity to the initial reviews, both by me and by the other reviewers. The additions to the manuscript, such as the sigmoid fits in figure 4 and supplemental analysis with restricted eccentricities, strengthen the paper. The motivation and discussion are now clearer as well.

I have only remaining suggestion (which should not require re-review by me). In response to my point 3 (about face perception at 40°), there is one other point the authors might make (if they agree with it). Specifically, the difference in retinotopy for the different face areas might predict that social judgments from face stimuli will decline less in performance as a function of eccentricity than identity judgments. It is nice to make behavioral predictions from novel neural measures where possible. In any case, just a thought.

Overall, interesting and compelling paper! Congratulations.

Jonathan Winawer

Thank you very much!

We appreciate your suggestion and agree that including such a behavioral prediction would improve the manuscript. Accordingly, we have added the following to the Discussion on page 15: *“A behavioral prediction from our data as well as prior research⁵¹, which can be tested in the future, is that performance on tasks related to the lateral stream (e.g., judging facial expressions) will decline less as a function of eccentricity than tasks related to the ventral stream (e.g., judging facial identity).”*

Reviewer #3 (Remarks to the Author):

The authors addressed my main concern. The revised manuscript includes data from additional participants in particular for mSTS. Results are now more convincing with respect to the differences in pRF between the ventral and lateral face areas.

Thank you, we are happy to hear that.

I have only a couple remaining comments:

1. Please add effect size and confidence intervals to all the reports of statistical effects.

In response to point 1, standard error and sample size, which together represent equivalent information to confidence intervals, were already included in the manuscript for all relevant tests. We have now added effect size estimates (η_p^2 for F tests and Cohen's d for t-tests) for all statistical tests in the main manuscript as requested.

2. I am wondering to what extent these findings are specific to face areas or are a general difference between ventral areas and the STS.

Our data suggest that our findings cannot be explained entirely by a general difference between ventral areas and the STS, as the ventral area which is selective to places: CoS-places resembles the data of lateral face-selective regions along the STS. However, there are other category selective regions in both the ventral and lateral streams, and the extent to which our observations extends to other regions is an interesting question for future research.

Therefore, we have added the following to the Discussion on page 14 to summarize the CoS-places findings and address this point: *“Furthermore, these properties do not appear to simply reflect a general difference between ventral and lateral visual cortex, as pRFs and white matter connections of CoS-places, which is in ventral temporal cortex, more closely resemble those of the lateral face-selective regions. However, an interesting question for future research is whether these differences across face-selective areas in the ventral and lateral streams extend to body-selective regions, which neighbor the face-selective regions in both streams^{46,47}.”*